# Coordinated repression of totipotency-associated gene loci by histone methyltransferase EHMT2 via LINE1 regulatory elements

Kaushiki Chatterjee [1,2], Christopher Mitsuo Uyehara [1,2], Kritika Kasliwal [1], Subhashini Madhuranath[1], Laurianne Scourzic [1], Alexander Polyzos [1], Effie Apostolou [1✉] & Matthias Stadtfeld [1✉]

## Abstract

**Mouse embryonic stem cells (mESCs), in addition to differentiating into the three germ layers, can reverse typical developmental trajectories, as exemplified by their ability to de-differentiate into 2-cell-like cells (2CLCs) that resemble the mammalian embryo during zygotic genome activation (ZGA). This unique property offers the opportunity to elucidate the molecular principles that govern the pre-implantation stages of mammalian development. Here, we dissect the functions of the chromatin repressor EHMT2, a candidate antagonist of the mESC-to-2CLC transition, by leveraging a multipurpose allele for acute protein depletion and efficient immunoprecipitation. Our experiments revealed distinct principles of EHMT2-mediated gene repression in mESCs based on specific chromatin binding patterns and protein co-factors. Most notably, EHMT2 directly represses large clusters of co-regulated gene loci that comprise a significant fraction of the 2CLC-specific transcriptome by initiating H3K9me2 spreading from distal LINE-1 elements. EHMT2 counteracts the recruitment of the activator DPPA2/4 to promoter-proximal endogenous retroviral elements (ERVs) at 2CLC genes. EHMT2 depletion enhances the expression of ZGA-associated transcripts in 2CLCs and synergizes with spliceosome inhibition and retinoic acid signaling to facilitate the mESC-to-2CLC transition. In contrast to ZGA-associated genes, the repression of germ layer-associated transcripts by EHMT2 occurs outside of gene clusters, in collaboration with ZFP462, and involves binding to non-repetitive candidate enhancers. Our observations provide novel mechanistic insight into how pluripotent cells achieve attenuation of their bidirectional differentiation potential and reveal unique transcriptional features of murine totipotent cells.**

**Keywords** H3K9 Methylation; EHMT2; Totipotency; Pluripotency; LINE-1
**Subject Categories** Chromatin, Transcription & Genomics; Stem Cells & Regenerative Medicine

## Introduction

Mammalian stem and progenitor cells exhibit unidirectional developmental plasticity and generally do not revert to earlier stages of differentiation. This is the basis for Waddington's epigenetic landscape (Baedke, 2013). Cultured naive pluripotent stem cells, such as mouse embryonic stem cells (mESC), are characterized by their ability to differentiate into all three germ layers and derivative tissues, reflecting the developmental potential of their in vivo counterpart, the epiblast of the pre-implantation blastocyst. However, mESCs can also spontaneously reverse physiological developmental trajectories and, at low frequencies, give rise to so-called 2-cell-like cells (2CLCs) (Macfarlan et al, 2012). 2CLCs express genes that are transiently activated during zygotic genome activation (ZGA), which in mice occurs at the two-cell stage before becoming silenced during later development (Macfarlan et al, 2012). Several transcriptional regulators, such as the transcription factors (TFs) DPPA2/4 (De Iaco et al, 2019; Eckersley-Maslin et al, 2019) and DUX (Hendrickson et al, 2017; Yang et al, 2020), have been reported to activate ZGA-associated genes during the mESC-to-2CLC transition. Select 2CLC-associated genes, such as *Zscan4* (Falco et al, 2007) and *Obox* (Ji et al, 2023), are organized in gene clusters, the regulatory logic of which remains unknown.

The unusual bidirectional potential of mESCs suggests that molecular mechanisms exist that not only support but also counteract "forward" (into germ layers) and "backward" (into 2CLC) differentiation, thereby allowing mESCs to self-renew in an undifferentiated state. For example, core pluripotency TFs, such as OCT4, can recruit repressive chromatin modifiers to loci encoding signaling and transcriptional regulators required for germ layer differentiation (Jerabek et al, 2014). Several distinct cellular pathways and regulators have been reported to be involved in regulating the mESC-to-2CLC conversion (Jia Yu and Guan, 2024; Nakatani and Torres-Padilla, 2023), suggesting the existence of multiple regulatory layers that converge on suppressing the unscheduled reactivation of 2CLC-associated transcripts in pluripotent cells. However, whether the same regulators are involved in counteracting 2CLC formation and forward differentiation of mESCs remains unexplored.

[1]Sanford I. Weill Department of Medicine, Sandra and Edward Meyer Cancer Center, Weill Cornell Medicine, New York, NY 10065, USA. [2]These authors contributed equally: Kaushiki Chatterjee, Christopher Mitsuo Uyehara. ✉E-mail: efa2001@med.cornell.edu; mas4011@med.cornell.edu

The extensive differences in genome accessibility (Hendrickson et al, 2017), histone mobility (Boskovic et al, 2014), chromatin marks (Eckersley-Maslin et al, 2016), and chromatin topology (Zhu et al, 2021) distinguishing mESCs and 2CLCs make epigenetic regulators prime candidates for modulating the interconversion between these cells. Accordingly, the inhibition of histone-modifying enzymes can increase the abundance of 2CLCs in mESCs cultures (Macfarlan et al, 2012). However, the specific target genes of these enzymes and the underlying regulatory mechanisms remain unknown.

Euchromatic histone methyltransferase 2 (EHMT2), also known as G9a, was identified as the enzyme catalyzing the repressive H3K9me2 mark in gene-rich regions outside of the pericentromeric heterochromatin of the mammalian genome (Rice et al, 2003; Tachibana et al, 2001) together with its dimeric interaction partner EHMT1 (Tachibana et al, 2005). EHMT2 null mice die during early organogenesis with multi-lineage defects (Tachibana et al, 2002), but transcriptional dysregulation in the absence of EHMT2 is already evident at pre-implantation stages (Zylicz et al, 2018). Since EHMT2 does not contain any DNA-binding domain, it is believed to gain target gene specificity by cell type-specific recruiting factors such as the TF ZFP462 in mESCs (Yelagandula et al, 2023). Cultures of mESCs deficient for EHMT2 exhibit upregulation of gene loci associated with neurodevelopment and other germ layers (Mozzetta et al, 2014), as well as ectopic activation of transposable elements (TE) highly expressed in two-cell embryos, such as ERVs (Maksakova et al, 2013) and an elevated number of 2CLCs (Macfarlan et al, 2012). Combined, these observations suggest a potential role for EHMT2 in counteracting both "forward" and "backward" differentiation in mESCs.

Here, we combine a novel degron allele with 2CLC reporters to explore the molecular role of EHMT2 in mouse ESCs. Acute EHMT2 depletion reveals that co-regulated gene clusters, which we term "EHMT2 coordinately repressed domains" (ECORDs), are a defining feature of 2CLCs, from which a significant fraction (~30%) of transcripts specific to these cells originate. Genes within ECORDS are highly expressed during ZGA in vivo, and we show that loss of EHMT2 further elevates ZGA-associated transcripts in 2CLCs. Mechanistically, spreading of EHMT2-catalyzed H3K9me2 domains from LINE-1 elements in mESCs prevents the binding of the activating transcription factor DPPA2/4 to ECORDs. At the same time, EHMT2 synergizes with ZFP462 to silence differentiation-associated genes in other genomic regions. Our study demonstrates that a single chromatin repressor can engage in distinct gene-regulatory modes to preserve the developmental plasticity of pluripotent stem cells, providing molecular insights into the underlying mechanism.

# Results

## Acute EHMT2 depletion de-represses distinct categories of gene loci in mESCs

To facilitate the study of EHMT2's gene-regulatory functions, we replaced its STOP codon in mouse embryonic stem cells (mESCs) with an in-frame transgenic cassette encoding the degron tag FKBP12$^{F36V}$ ("dTAG") (Nabet et al, 2018) and two copies of the hemagglutinin (HA) tag (Fig. 1A). The degron design also contains a mCherry reporter to capture changes in the transcriptional activity of Ehmt2 and to facilitate isolation of correctly targeted cells (Fig. 1A). We generated several PCR-validated homozygous EHMT2-dTAG mESC lines. Flow cytometric analysis of the mCherry reporter confirmed homogeneous Ehmt2 expression in pluripotent cells (Appendix Fig. S1A). Culture of EHMT2-dTAG mESCs for 24 hours (h) in the presence of the degrader dTAG-13 (Nabet et al, 2018) resulted in near complete elimination of EHMT2 protein as measured by Western Blot (Appendix Fig. S1B). Quantification by flow cytometry revealed that total EHMT2 depletion was achieved after six hours of dTAG-13 treatment, followed by a delayed reduction in the levels of the H3K9me2 mark, which is catalyzed by the EHMT1:EHMT2 complex (Fig. 1B; Appendix Fig. S1C). No reduction of H3K9me2 levels was observed in EHMT2-dTAG mESCs in the absence of dTAG-13 (Appendix Fig. S1D). Thus, our transgenic system achieves robust, dTAG-13-dependent control of EHMT2 levels in mESCs.

To determine the transcriptional consequences of acute EHMT2 depletion, we conducted RNA-sequencing (RNA-seq) experiments with three independent EHMT2-dTAG cell clones 24 h after dTAG-13 administration (Appendix Fig. S1E). This revealed 631 differentially expressed genes (DEGs) (abs(log2FC)>1; P-adj <0.05) compared to DMSO-treated controls (Fig. 1C). Consistent with a predominant role of EHMT2 as a transcriptional repressor in mESCs (Shinkai and Tachibana, 2011; Tachibana et al, 2008), most DEGs (446/631 or 70.7%) were upregulated and upregulated DEGs also had higher fold changes than downregulated DEGs (Fig. 1C). To determine the longer-term consequences of EHMT degradation, we also conducted RNA-seq 7 days (d) after continuous dTAG-13 administration (Appendix Fig. S1E). Although mESCs treated in this manner remained viable, retained an undifferentiated morphology, and exhibited no overt growth defect, 7 days RNA-seq revealed a substantially more pronounced transcriptional effect of EHMT2 loss with a total of 1615 DEGs, a slight majority being upregulated (58.0% or 936/1615 genes) (Fig. 1D) (Dataset EV1). Previous work has reported that EHMT2 antagonizes the expression of specific endogenous retroviral transcripts (Maksakova et al, 2013). In line with this, EHMT2 depletion resulted in dysregulation of several repeat elements with upregulation of specific ERVK/ERVL LTR families among the earliest and most pronounced consequences (Appendix Fig. S1F). At 7 days, we observed further derepression of ERVs, which is in line with a significant role of EHMT2 in stably repressing these repeat families (Appendix Fig. S1G). Most DEGs upregulated ("DEG$^{UP}$") upon prolonged EHMT2 depletion had already shown at least a trend toward upregulation after acute EHMT2 depletion (Appendix Fig. S1H). However, we also observed subsets of DEG$^{UP}$ that were specific to the 24 h (C5 in Appendix Fig. S1H) or 7 days (C1 and C6) timepoint, possibly suggesting the existence of compensatory repressive mechanisms or the accumulation of indirect molecular effects of EHMT2 loss, respectively.

A striking outcome of our RNA-seq analysis was that a subset of DEG$^{UP}$ genes was in linear proximity to one another, forming apparent clusters of genes that showed a similar response to EHMT2 depletion. By projecting the chromosomal locations and fold changes of 24 h and 7 days DEGs, we confirmed that many strongly upregulated (but not downregulated) genes were organized into clusters along the linear genome, suggesting coordinated repression by EHMT2 (Fig. 1E). To identify clusters in an unbiased

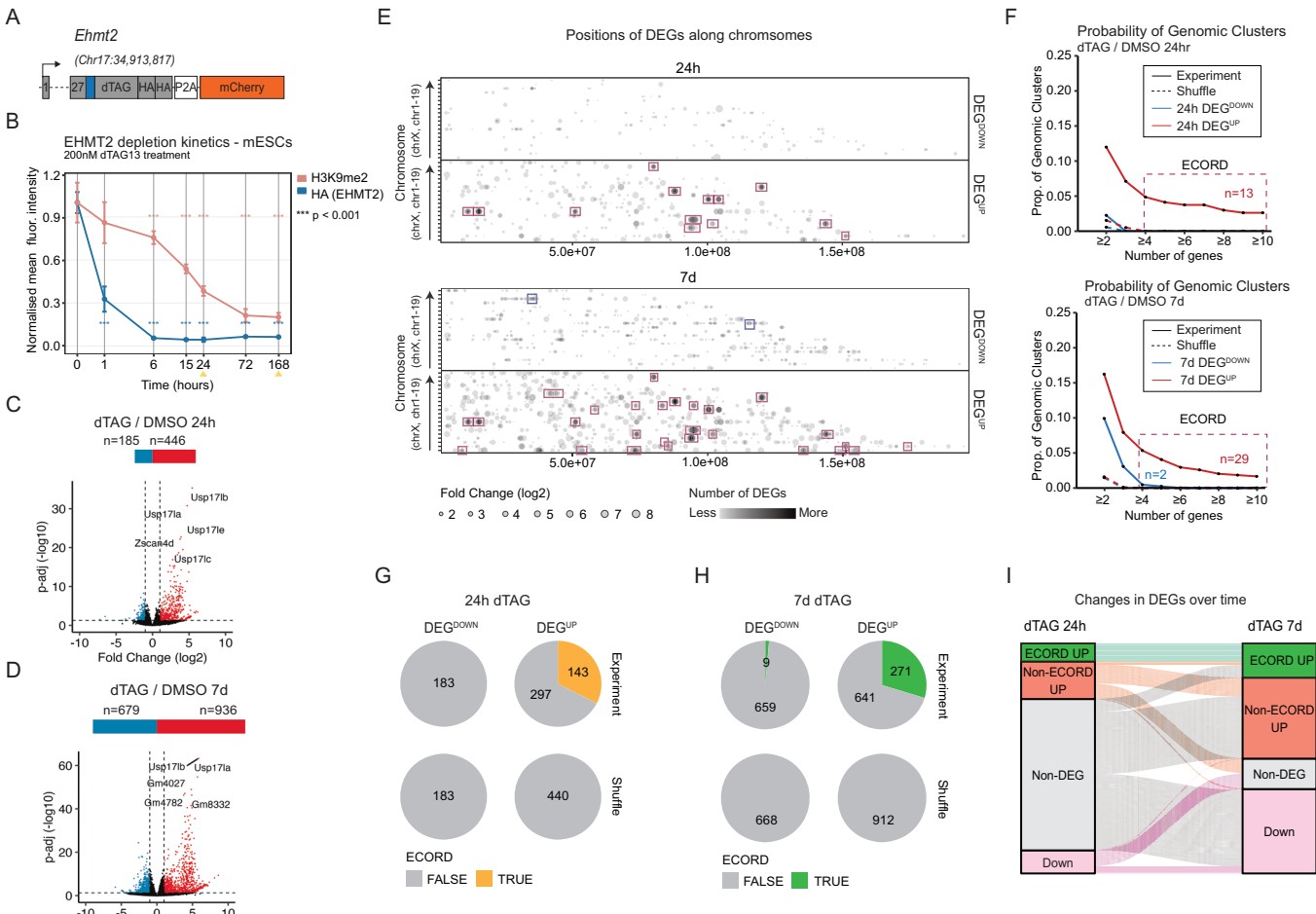

**Figure 1. Acute EHMT2 depletion reveals distinct categories of target genes in mESCs.**

(A) Schematic of the EHMT2 degron allele. A degron (dTAG), two HA tags, and a mCherry transcriptional reporter were integrated in-frame at the single STOP codon of the endogenous *Ehmt2* locus. (B) Flow cytometry analysis of EHMT2 (HA) and H3K9me2 levels over time in response to continuous dTAG treatment. Yellow arrowheads indicate the isolation of samples for RNA-seq analysis. ***$P < 0.001$; unpaired $t$ test. Error bars represent the mean with SD ($n = 3$ technical replicates from two independent cell lines). Exact $P$ values are listed in Dataset EV6. (C, D) Volcano plots showing numbers of differentially expressed genes (DEGs) following 24 h and 7 days of continuous dTAG treatment, respectively. DEG: Adj. $P$ value < 0.05 and absolute log2 fold change ≥1, Wald test using DESeq2. The top 5 DEGs by fold change are highlighted. Three biological replicates were used. (E) Positions of 24 h and 7 days EHMT2 DEGs along the linear genome. The opacity of the points was decreased so areas with large numbers of high FC DEGs appear darker. Boxes highlight genes (red: upregulated; blue: downregulated) that fall into EHMT2 Coordinately Repressed Domains (ECORDs) as defined in (F) and Appendix Fig. S1H,I. (F) Probability curves showing the fraction of gene clusters with ≥$n$ DEGs and <50% static genes that do not break TAD boundaries (see "Results" and "Methods" sections). For subsequent analyses, "ECORDs" were defined as clusters with $n ≥ 4$ DEGs and <0.5 proportion of static genes. (G, H) Fraction of DEGs that fall into ECORDs at 24 h and 7 days, respectively. (F, G) "Shuffle" refers to a random sample of genes expressed in either DMSO or dTAG conditions. (I) Alluvial plot showing changes in DEGs between 24 h and 7 d. Source data are available online for this figure.

and quantitative fashion, we counted the number of DEGs that occurred in sequence along the linear genome. The cluster was interrupted if (1) the next DEG changed in the opposite direction (*P*-adj <0.05, no fold-cutoff), or (2) the cluster crossed a TAD boundary (Di Giammartino et al, 2019). This analysis revealed that DEG^UP clusters spanned larger genomic regions than DEG^DOWN clusters and contained a significantly lower proportion of static (expressed and *P*-adjusted >0.05) genes (Fig. 1F; Appendix Fig. S1I,J). In some cases, >20 DEG^UP occurred in sequence with only 1–2 static genes. This analysis confirmed that clustering is a feature unique to a subset of DEG^UP that does not happen to DEG^DOWN or through chance. We will refer to genomic clusters with

≥4 DEG^UP and <50% static genes as EHMT2 Coordinately Repressed Domains or ECORDs. Overall, we identified 13 ECORDs at 24 h and 29 ECORDs at 7 d (Fig. 1F; Appendix Fig. S1K), which comprised 32.1% and 29.0% of DEG^UP, respectively, at these two time points (Fig. 1G,H) (Dataset EV2). All 24 h ECORDs were maintained at 7 days (Fig. 1I), and the vast majority (24 out of 29) of 7 days ECORDs had at least one DEG^UP at 24 h, demonstrating that EHMT2 loss causes early and sustained upregulation of genes within ECORDs. In contrast, some 24 h DEG^UP outside of ECORDs ("non-ECORD DEGs") were no longer identified as DEG^UP at 7 days, possibly suggesting the existence of compensatory repressive mechanisms at non-clustered gene loci (Fig. 1I).

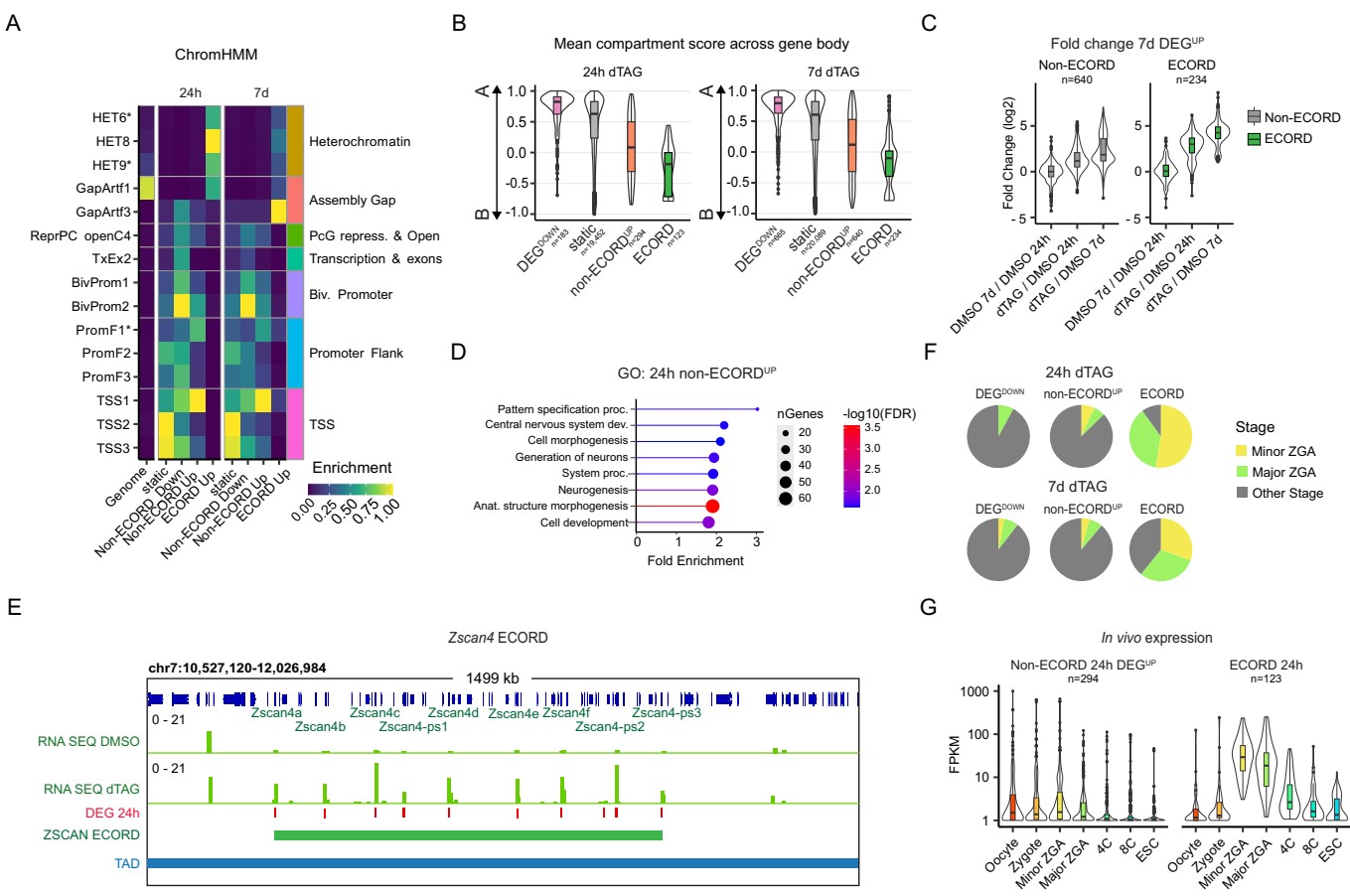

**Figure 2. Genes in ECORDs have distinct properties and are enriched for genes involved in Zygotic Genome Activation (ZGA).**

(A) ChromHMM of promoters of ECORD DEGs using a 100-state model (Vu and Ernst, 2023). The descriptions of starred terms are provided in Fig.S2A. (B) Mean compartment strength over different categories of genes following 24 h and 7 days of dTAG treatment. Compartment scores were obtained from GSE113431 (Di Giammartino et al, 2019). All expressed genes ($n = 19,635$ for 24 h and $n = 20,754$ for 7 days). (C) Fold-changes (log2) of DEGs upregulated after 7 days of dTAG treatment ($n = 874$ genes), split by whether the DEG is within an ECORD. (D) Gene Ontology (GO) terms of genes upregulated after 24 h dTAG treatment for Non-ECORD DEGs. (E) Browser shot of RNA levels across the ECORD encompassing the *Zscan4* genes, canonical markers. TAD boundaries were obtained from GSE113339 (Di Giammartino et al, 2019). (F) Overlap of 24 h DEGs with gene categories from an in vivo time course of pre-implantation embryonic development (Wu et al, 2016). Only genes that overlapped annotated categories were included. (G) Expression of 24 h DEGs at the indicated stages of in vivo development and mESCs. Only genes with an FPKM $\geq 10$ in at least one sample type were included ($n = 417$ genes). In the boxplots in (B, C, G), the center line marks the median; the box hinge ranges mark the upper and lower quartiles; the upper and lower whiskers mark the lowest value within 1.5 times the interquartile range (IQR). Points outside 1.5 times the IQR are indicated as outlier points.

## ECORDs are characterized by repressive chromatin and are associated with zygotic genome activation

The results described so far show that genes repressed by EHMT2 in mESC cultures can be separated into two broad groups based on whether they reside within or outside ECORDs. To elucidate potential regulatory differences between these groups, we analyzed their chromatin states with ChromHMM (Ernst and Kellis, 2017; Vu and Ernst, 2023). ECORD genes are strongly enriched for heterochromatin features such as H3K9me3, transposable elements (including LTRs and LINE-1 elements), and for the "Assembly Gap" class, which is characteristic of repetitive DNA (Fig. 2A; Appendix Fig. S2A; Dataset EV3). In contrast, non-ECORD DEG[UP] and DEG[DOWN] enriched for chromatin features associated with euchromatin and active transcription (Fig. 2A; Appendix Fig. S2A; Dataset EV3). Accordingly, analysis of public Hi-C data (Di

Giammartino et al, 2019) showed that ECORDs preferentially localized within (inactive) B compartments (Lieberman-Aiden et al, 2009) while non-ECORD DEG[UP] showed a weak enrichment for the A compartments at both 24 h and 7 days (Fig. 2B). Collectively, these observations suggest that ECORD DEGs represent a more repressed ground state than non-ECORD DEG[UP] in mESCs. In agreement, ECORD DEG[UP] showed a more substantial degree of upregulation upon EHMT2 loss than non-ECORD DEG[UP] (Fig. 2C). Importantly, the expression of almost all ECORD DEGs—as well as non-ECORD DEGs—reverted to physiological levels upon dTAG-13 washout and EHMT2 recovery (Appendix Fig. S2B–D), demonstrating that EHMT2 is directly responsible for the repression of these loci and can regain transcriptional control after being transiently depleted.

To understand the potential biological relevance of genes repressed by EHMT2 in mESCs, we performed a Gene Ontology

(GO) analysis. This showed that non-ECORD 24 h DEG$^{UP}$ were associated with developmental processes such as morphogenesis, neurogenesis, and organ development (Fig. 2D; Dataset EV5), which is consistent with the notion that EHMT2 functions in pluripotent cells to repress the premature expression of genes with regulatory roles during the post-implantation stages of development. This finding aligns with the embryonic lethality of EHMT2 KO mice during organogenesis (Tachibana et al, 2002; Tachibana et al, 2005). In contrast, ECORD DEGs were not associated with post-implantation development but were enriched for regulators of RNA localization and nuclear transport (Appendix Fig. S2E; Dataset EV5). We observed that ECORDs included genes known to become activated during zygotic genome activation (ZGA), such as *Zscan4* (Falco et al, 2007) (Fig. 2E) and *Obox* (Ji et al, 2023) loci. To further explore this, we compared our DEGs to a published dataset that characterized stage-specific transcripts during early mouse embryogenesis in vivo (Wu et al, 2016). Indeed, both 24 h and 7 days ECORD DEG$^{UP}$ strongly overlapped with ZGA-associated transcripts, which in mouse embryos are expressed at the highest levels during the early 2-cell (minor ZGA) stage, concomitant with the onset of ZGA (Fig. 2F,G). Non-ECORD DEG$^{UP}$ did not exhibit this pattern (Fig. 2G).

Together, these results support the notion that EHMT2 represses at least two broadly distinct categories of target gene loci in mouse pluripotent cells: clustered ECORD genes within heterochromatic regions, which are transiently activated during ZGA, and non-clustered, euchromatic loci encoding genes involved in later developmental stages.

## EHMT2 limits the entry rate of mESCs into a 2-cell-like transcriptional state in collaboration with other pathways

Cultures of mESCs can contain a small percentage of cells in a transient, 2-cell-like (2CLC) state characterized by the high-level expression of ZGA-associated genes such as *Zscan4* (Jia Yu and Guan, 2024; Nakatani and Torres-Padilla, 2023). Therefore, we hypothesized that the observed widespread upregulation of ZGA genes organized in ECORDs might reflect a change in the composition of our cell cultures, resulting in a higher percentage of 2CLCs. To enable dissecting the role of EHMT2 in controlling the emergence of 2CLCs and, ultimately, the control of ZGA-associated transcription, we generated EHMT2-dTAG mESC lines carrying destabilized, fast-folding TurboGFP reporters driven from murine endogenous retrovirus-L (MERVL) promoter elements (Ishiuchi et al, 2015) ("MERVL-GFP mESCs") (Fig. 3A). The activation of MERVL repeats is a hallmark of ZGA whose visualization is an established approach to identifying 2CLCs in culture (Jia et al, 2024; Nakatani and Torres-Padilla, 2023). Furthermore, EHMT2 has been shown to repress MERVL elements in mESCs (Maksakova et al, 2013), a finding confirmed by our RNA-seq analysis of bulk cultures (see Appendix Fig. S1F,G). MERVL-GFP$^+$ cells expressed the ZGA-associated, ECORD-encoded (Fig. 2E) transcription factor (TF) ZSCAN4 (Fig. 3B; Appendix Fig. S3A) and exhibited strongly reduced levels of the pluripotency-associated surface markers SSEA-1 and EpCAM (Polo et al, 2012) (Fig. 3C). Hereafter, we refer to MERVL-GFP$^+$ cells as "2CLCs" and to MERVL-GFP$^-$ cells as "mESCs".

EHMT2 depletion for 24 h resulted in a significant increase (>2.5 fold) in the percentage of 2CLCs compared to DMSO cultures (7–8% in dTAG vs 2–3% in DMSO) (Fig. 3D,E). The increased abundance of 2CLCs in cultures treated with dTAG-13 reached statistical significance after 12 h (Appendix Fig. S3B) and continued to grow until 3 days, after which the abundance stalled concomitant with cell passaging (Appendix Fig. S3C). These observations suggest that acute EHMT2 depletion in mESCs facilitates entry into the 2CLC state. Still, cells do not continue to accumulate in this state due to an apparent growth disadvantage of 2CLCs in standard mESC culture conditions.

Recent studies have revealed several distinct cellular pathways whose manipulation can increase the abundance of 2CLCs in mESC cultures, including the inhibition of spliceosome activity (Shen et al, 2021) and retinoic acid receptor (RAR) signaling (Iturbide et al, 2021). To determine the functional interplay of these pathways with EHMT2 in modulating the mESC-to-2CLC transition, we depleted EHMT2 in MERVL-GFP mESC cultures in the presence or absence of the spliceosome inhibitor Pladienolide B (PB) or the RAR agonist TTNBP (TT). Treatment with either compound alone significantly increased the proportion of 2CLCs compared to DMSO controls (Appendix Fig. S3D), supporting the aforementioned prior findings. Concomitant administration of dTAG-13 further increased the percentage of 2CLCs in both instances (Appendix Fig. S3D), suggesting that EHMT2 activity is not affected by either treatment. We did not observe an increase in 2CLCs after treatment with two other compounds—the GSK3 inhibitor 1-Azakenpaullone and the kinase blocker WS6—used to establish cultures of cells resembling 2-cell embryos (Appendix Fig. S3D), suggesting that these compounds may not operate by facilitating the initial mESC-to-2CLC transition. While EHMT2 depletion and RAR activation were associated with minor changes in overall cell numbers, visual inspection suggested that spliceosome inhibition significantly reduced the number of viable cells in mESC cultures. This observation suggests that adverse selection driven by distinct metabolic requirements between mESCs and 2CLCs might partly explain the increased ratio of 2CLC cells observed upon spliceosome inhibition. In contrast, our results support that EHMT2 depletion increases the proportion of 2CLC cells in culture by facilitating the transition into the 2CLC state through a mechanism that is at least partially distinct from both retinoic acid signaling and spliceosome inhibition.

## EHMT2 has distinct gene-regulatory functions in 2CLCs and mESCs

The increased abundance of 2CLCs following EHMT2 depletion could explain the apparent upregulation of ECORDs and other ZGA-associated genes observed in the RNA-seq of bulk mESC cultures. However, our analysis cannot exclude that EHMT2 depletion introduces additional transcriptional changes in 2CLCs or mESCs. In addition, bulk cell analysis likely underestimates the actual number of ECORDs in the genome. To address these limitations, we used our MERVL-GFP/EHMT2-dTAG system to isolate highly pure (>95%) populations of 2CLCs and mESCs three days (3 days) after EHMT2 depletion (Appendix Fig. S3E) and performed RNA-seq (Appendix Fig. S3F). The comparison of DMSO-treated mESCs and 2CLCs revealed 3314 2CLC-associated transcripts and 1417 mESC-associated transcripts. A substantial

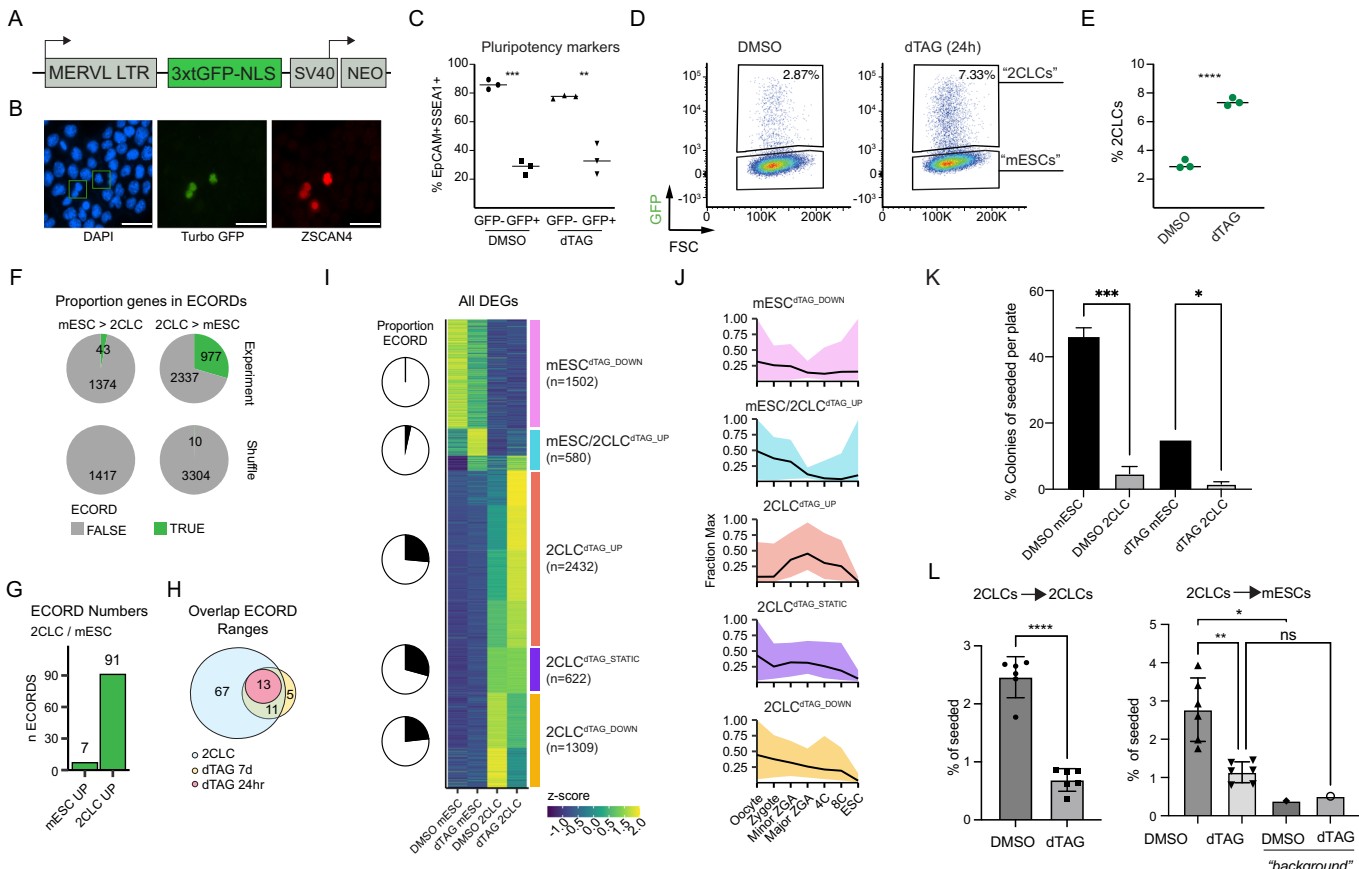

**Figure 3. EHMT2 acts as a gatekeeper for entry into the 2-cell-like state.**

(A) Simplified schematic of the MERVL reporter construct used to generate MERVL-GFP mESCs. (B) Immunofluorescence image of a representative field showing co-expression of MERVL-GFP (green) and ZSCAN4 (red). DAPI staining is indicated in blue. Green boxes indicate co-expressing cells. (C) Percent co-expression of pluripotency-associated surface markers EpCAM and SSEA1 in mESC and 2CLC states after treatment with DMSO or dTAG, as measured by flow cytometry. **$P < 0.01$, ****$P < 0.0001$; unpaired $t$ tests. $N = 3$ biological replicates. (D) FACS analysis of MERVL-GFP mESCs following 24 h DMSO or dTAG treatment. MERVL-GFP⁺ cells are referred to as "2-cell-like-cells (2CLC)" and MERVL GFP⁻ cells as "mESCs". (E) Percentage of 2CLCs identified by flow cytometry after 24 h dTAG or DMSO treatment ($n = 3$ replicates). ****$P < 0.0001$; unpaired $t$ test. $N = 3$ biological replicates. (F) Numbers of mESC-associated and 2CLC-associated DEGs in DMSO conditions that fall into ECORDs. (G) Number of ECORDs in DMSO mESCs and 2CLC. (H) Overlap of genomic ranges for ECORDs identified in bulk RNAseq (dTAG > DMSO) and ECORDs identified in sorted RNAseq (DMSO 2CLC > mESC). (I) K-medoids clustering of all 2CLC- and mESC-associated DEGs. The signal is z-scored by row. The proportion of genes in ECORDs is indicated for each cluster. (J) Expression (fraction max) trends of DEGs in different clusters during pre-implantation development (Wu et al, 2016). Only genes with an FPKM ≥ 10 in at least one sample type were included. The line indicates the median, while the upper and lower bounds of ribbons indicate the 25th and 75th percentiles. (K) Percentage of colonies formed per 96-well plate. Sorted cells were cultured in treatment media (DMSO or dTAG). *$P < 0.05$, ***$P < 0.001$; one-way ANOVA. Error bars represent the mean with SD ($N = 2$ biological replicates). (L) Quantification of the indicated lineage output of sorted 2CLCs derived in different media conditions after 24 h. *$P < 0.05$, **$P < 0.01$, ****$P < 0.0001$; unpaired $t$ test (left panel) and one-way ANOVA (right panel). Error bars represent the mean with SD. $N = 6$ biological replicates. "Background" indicates the abundance of mESCs expected to arise due to the low-level impurities of sorted 2CLCs (see Appendix Fig. S3E). Source data are available online for this figure.

proportion of 2CLC-associated genes were organized in ECORDs (977 of 3314 or 29.5%). In comparison, only 3% (43 of 1417) of mESC-enriched transcripts were clustered (Fig. 3F). Specifically, we identified a total of 91 ECORDs in 2CLCs (mean size 10.7 genes, range 5–66) but only seven ECORDs in mESCs (mean size 6.1 genes; range 5–8) (Fig. 3G; Appendix Fig. S3G,H) (Dataset EV2). The majority of ECORDs we had detected in bulk RNA-seq (13/13 24 h and 24/29 7 days after EHMT2 depletion, respectively) overlapped with 2CLC-specific ECORDs (Fig. 3H), demonstrating that ECORDs are a feature of 2CLCs but less so of mESCs. Of note, 2CLC-associated transcripts within ECORDs showed more robust differential expression between 2CLC and mESC and higher absolute expression levels in 2CLCs compared to non-ECORD

genes (Appendix Fig. S3I,J), further supporting a strong association of ECORD activation and 2CLC identity. Of note, strong upregulation of 2CLC-associated transcripts in ECORDs, but not of transcripts outside of ECORDs, was evident in dTAG-treated purified mESCs (Appendix Fig. S3K), suggesting that derepression of ECORDs is an early event during entry into the 2CLC state driven by EHMT2 loss. Together, these observations establish that the coordinated activation of gene clusters is a defining and widespread feature of gene expression in both spontaneously arising 2CLCs and 2CLCs triggered by EHMT2 depletion.

To further characterize the impact of EHMT2 depletion on cell state-specific gene expression, we compared the transcriptome of purified mESCs and 2CLCs under both DMSO and dTAG-13

conditions. K-medoid clustering of all genes differentially expressed in at least one pairwise comparison ($n = 5784$ genes) defined five larger gene groups with distinct trends of transcriptional change in response to EHMT2 loss. Most prominently, we observed a large group of 2CLC-associated genes that were further upregulated in 2CLCs upon dTAG-13 treatment, many of them strongly ("2CLC$^{dTAG\_UP}$") (Fig. 3I). Smaller groups of 2CLC-associated transcripts were either weakly downregulated ("2CLC$^{dTAG\_DOWN}$") or remained unaffected ("2CLC$^{dTAG\_STATIC}$"). All three groups of 2CLC-associated DEGs showed a similar enrichment for ECORDs (~25% of genes) (Fig. 3I), suggesting subtle differences in the transcriptional regulation of ECORDs downstream of EHMT2 depletion. We further observed a group of mESC-associated genes that were weakly downregulated upon EHMT2 depletion ("mESC$^{dTAG\_DOWN}$") and a group of genes upregulated in both cell types, albeit stronger in mESCs ("mESC/2CLC$^{dTAG\_UP}$") (Fig. 3I). Neither of these two groups showed enrichment for ECORDs, with those assigned to mESC/2CLC$^{dTAG\_UP}$ representing the rare ECORDs comprised of mESC-associated genes (Dataset EV2). These results show that, in addition to facilitating the mESC-to-2CLC transition, EHMT2 depletion also affects the gene expression of both mESCs and 2CLCs.

To gauge the potential biological relevance of the impact of EHMT2 depletion on 2CLC gene expression beyond facilitating the mESCs-to-2CLCs transition, we determined the expression kinetics of our gene groups in early mouse embryos and mESCs using published in vivo RNA-seq data (Wu et al, 2016). This revealed that 2CLC$^{dTAG\_UP}$ DEGs, but neither of the other two 2CLC-associated gene groups, were upregulated explicitly in 2-cell embryos at the time of ZGA (Fig. 3J). Both groups of mESC-associated DEGs showed no apparent upregulation at any stage assessed, consistent with the notion that they predominantly comprise genes expressed at later stages of development (Fig. 3J). This analysis suggests that EHMT2 depletion, in addition to facilitating the conversion of mESC into the 2CLC state, further solidifies a transcriptional program in 2CLCs that more closely resembles the developmental stage of ZGA.

To determine how EHMT2 depletion might impact the documented but poorly characterized ability of 2CLCs to return to a naive pluripotent state (Macfarlan et al, 2012), we conducted single-cell seeding experiments using purified 2CLCs and mESCs 72 h after EHMT2 depletion, as well as control cells. Sorted cells were allowed to grow in their respective treatment media. Quantification 5 days later revealed occasional mESCs colonies after seeding DMSO-treated 2CLCs, supporting the idea that these cells can revert to a pluripotent state in the presence of EHMT2 at low efficiency (Fig. 3K). In contrast, we observed virtually no colonies with naive pluripotent cell morphology in wells seeded with 2CLCs lacking EHMT2 (Fig. 3K). We confirmed this observation with 2CLCs that experienced EHMT2 depletion, but then were cultured without dTAG (Appendix Fig. S3L). We also observed a reduction in the seeding efficiency of mESCs exposed to dTAG-13 compared to control mESCs (Fig. 3K), possibly reflecting impaired self-renewal of mESCs caused by an increased propensity of these cells to transit into 2CLCs. To further determine the fate of 2CLCs, we re-analyzed the purified cells 24 h after sorting (Fig. 3L; Appendix Fig. S3M). This revealed that EHMT2-depleted 2CLCs generated even fewer progeny— both 2CLCs and mESCs—than spontaneously forming 2CLCs (Fig. 3L). In line with their overall

poor retention in culture, both dTAG- and DMSO-treated 2CLCs exhibited significantly elevated levels of the apoptosis marker Annexin V compared to mESCs (Appendix Fig. S3N). Together, these observations suggest that EHMT2-depleted 2CLCs do not stably retain a 2CLC identity—at least under standard mESC conditions—but exhibit evidence of elevated molecular stress that may result in cell death via a non-apoptotic mechanism and counteract the reversal of 2CLCs to a mESC state.

## EHMT2 genome occupancy in mESCs occurs at TEs within H3K9me2 domains

Our transcriptional profiling has shown that EHMT2 represses genes both inside ECORDs (sensitive in 2CLCs) and outside of ECORDs (sensitive in 2CLCs or mESCs) (Fig. 3I). To understand if EHMT2 regulates these genes directly, we conducted ChIP-seq experiments in EHMT2::dTAG mESCs ($n = 2$ lines) with antibodies against the HA tag incorporated in our degron allele (see Fig. 1A). Unlike most TFs, chromatin regulators such as EHMT2 do not directly engage with DNA, complicating reliable pulldown during ChIP-seq. Therefore, we also applied anti-HA ChIP-exo (Rossi Lai and Pugh, 2018), an alternative method for mapping the genome occupancy of transcriptional regulators at high resolution, spanning both euchromatic and heterochromatic regions (Skene and Henikoff, 2017). ChIP-seq and ChIP-exo replicates clustered together, demonstrating the reproducibility of our results and suggesting differences between the two assays (Appendix Fig. S4A). For downstream analysis, we implemented an alignment strategy that utilizes the STAR aligner to assign multimapping reads to their optimal genomic location (Teissandier et al, 2019). STAR increased the proportion of reads over genomic repeats, particularly for ChIP-exo (Appendix Fig. S4B), enabling the identification of peaks that would have otherwise been missed (Appendix Fig. S4C,D).

In total, we detected 12,266 EHMT2 peaks with ChIP-seq and 13,675 peaks with ChIP-exo. We observed that only a minority (~20%) of EHMT2 peaks were identified with both methods (Appendix Fig. S4E) (Dataset EV4). While the reasons for this discrepancy require further investigation, we surmise that the high signal-to-noise ratio of ChIP-exo (Rhee and Pugh, 2011) may allow the detection of weak binding events. At the same time, the more extensive fixation during ChIP-seq may stabilize EHMT2 binding away from DNA. To probe for evidence of EHMT2 activity in the vicinity of EHMT2 peaks, we conducted Ultra Low Input Native ChIP-seq (ULI-NChIP) (Brind'Amour et al, 2015; Brind'Amour and Lorincz, 2022) against the H3K9me2 mark in bulk mESC cultures. This revealed enrichment of H3K9me2 around the EHMT2 peak summits identified by either method, with the signal strengths in H3K9me2 and EHMT2 datasets generally correlating with one another (Appendix Fig. S4E). We also conducted HA ChIP-seq 24 h after EHMT2 depletion, which resulted in significantly reduced signal intensity across all peaks detected in cells expressing EHMT2 (Appendix Fig. S4E). Together, these observations support the reliability of our genome occupancy data and suggest that a combination of orthogonal ChIP methods might be required to detect the entire repertoire of target sites bound by chromatin regulators.

A high percentage of EHMT2 binding occurred at transposable elements (Fig. 4A), with EHMT2 peaks detected with both ChIP-exo and ChIP-seq approaching the binding frequency (>90%)

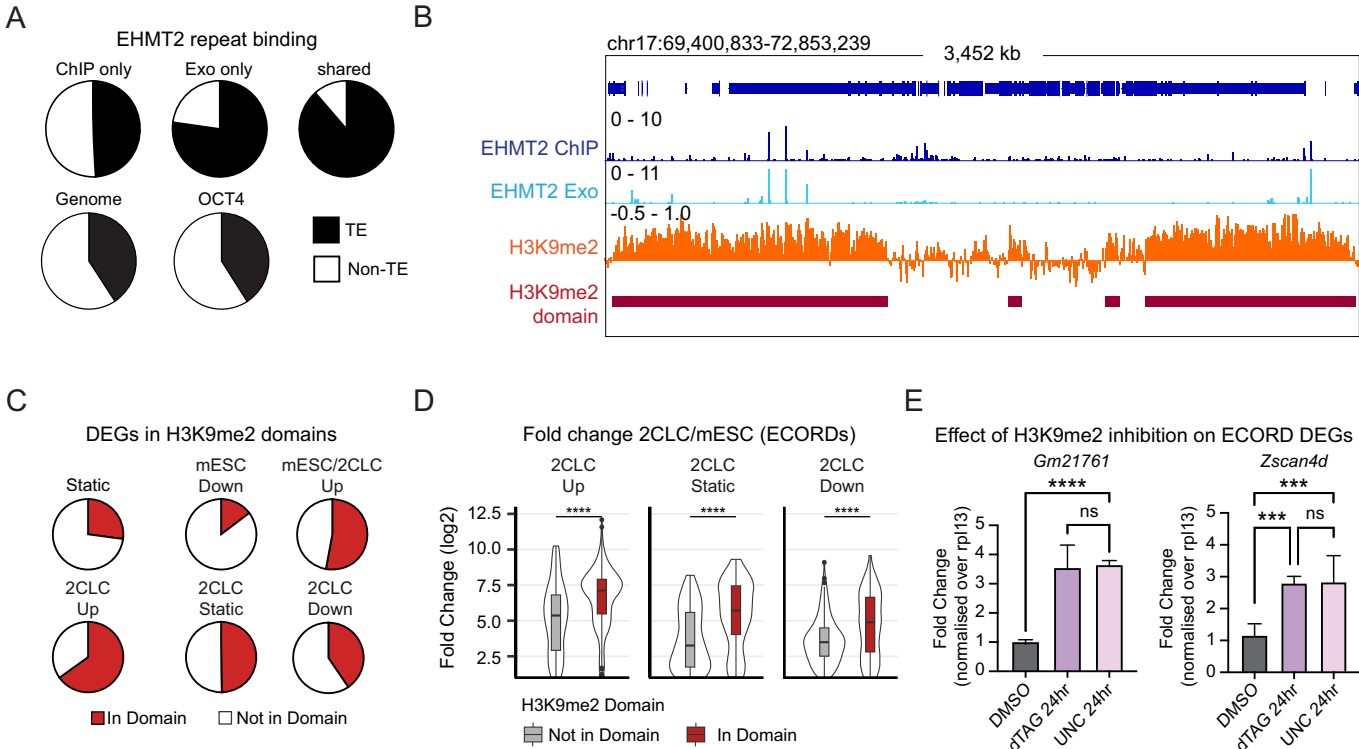

**Figure 4. H3K9me2 domains repress ECORD expression.**

(A) Proportion of indicated ChIP peaks that overlap repetitive (TE) elements. Proportion was calculated at the base pair level. (B) Example of EHMT2 binding and H3K9me2 signal inside and outside H3K9me2 domains. (C) Proportion of indicated DEG clusters (2CLC vs mESC) and control genes ("static") that overlap H3K9me2 domains. (D) Log2 Fold Changes (LFC) of DEGs in ECORDs split by whether they overlap an H3K9me2 domain. ****$P < 0.0001$; unpaired, two-sided, Wilcoxon rank-sum test with Bonferroni correction. Boxplots are defined in Fig. 2. (E) qPCR of the ECORD genes *Gm21761* and *Zscan4d* after treatment with the H3K9me2 inhibitor UNC0638. **$P < 0.01$, ****$P < 0.0001$; one-way ANOVA. Error bars represent the mean with SD. $N = 3$ biological replicates measured in duplicate. Exact $P$ values for (D, E) are listed in Dataset EV6. Source data are available online for this figure.

observed for TRIM28, a key repressor of endogenous retroviruses in mESCs (Rowe et al, 2010) (Fig. 4A; Appendix Fig. S4F). EHMT2 binding to repeats significantly surpassed the numbers expected by chance or observed for pluripotency-associated transcription factors such as OCT4 and KLF4, or the reported EHMT2 recruiters WIZ (Bian Chen and Yu, 2015) and ZFP462 (Yelagandula et al, 2023) by ChIP-seq processed through the same pipeline (Fig. 4A; Appendix Fig. S4F). EHMT2 binding was widespread at LINE elements (for shared and Exo-specific peaks) and LTR elements (for ChIP-specific peaks) (Appendix Fig. S4G), suggesting a possible involvement of these elements in recruiting EHMT2 to specific gene-regulatory circuits.

EHMT2 can repress genes by nucleating heterochromatin domains that spread across the linear genome until TAD boundaries limit their expansion (Fukuda et al, 2021; Yan et al, 2020). Consequently, individual EHMT2 binding sites do not necessarily need to overlap their transcriptional targets to exert their repressive role. To investigate the relationship between EHMT2 binding and gene expression, we called H3K9me2 domains using our ULI-NChIP data, which documented that most EHMT2 binding sites were found within H3K9me2 domains (Fig. 4B; Appendix Fig. S4H). When integrating RNA-seq and ULI-NChIP-seq data, we observed that genes upregulated upon EHMT2 depletion were more commonly located in H3K9me2 domains

compared to static genes or genes that were downregulated (Fig. 4C). This was true of both 2CLC-associated and mESC-associated genes (Fig. 4C) but was particularly evident for 2CLC$^{dTAG\_UP}$ DEGs in ECORDs (Appendix Fig. S4I). In addition, 2CLC-associated genes within H3K9me2 domains experienced more pronounced upregulation during the mESC-to-2CLC transition (Fig. 4D) and were also upregulated upon enzymatic inhibition of EHMT2 (Fig. 4E). EHMT2 inhibition and depletion resulted in similar increases in the abundance of 2CLCs, supporting the significance of the enzymatic activity of EHMT2 in this context (Appendix Fig. S4J). These observations underscore the importance of EHMT2-regulated H3K9me2 domains for ECORD repression and demonstrate that the enzymatic activity of EHMT2 is key to its role in suppressing the mESC-to-2CLC transition. Overall, these observations support that the repressive function of the observed H3K9me2 domains is dependent on EHMT2 and show that this repression is overcome when cells enter the 2CLC state. In addition, they suggest a possible role of TEs—particularly LINEs and LTRs—in EHMT2-mediated gene repression and the establishment of H3K9me2 domains in specific genomic regions.

To gain insight into potential distinct regulatory mechanisms of EHMT2 inside and outside of ECORDs, as well as to identify possible EHMT2 co-factors or antagonists, we analyzed our combined EHMT2 ChIP peaks together with published and in-house ChIP-seq data of

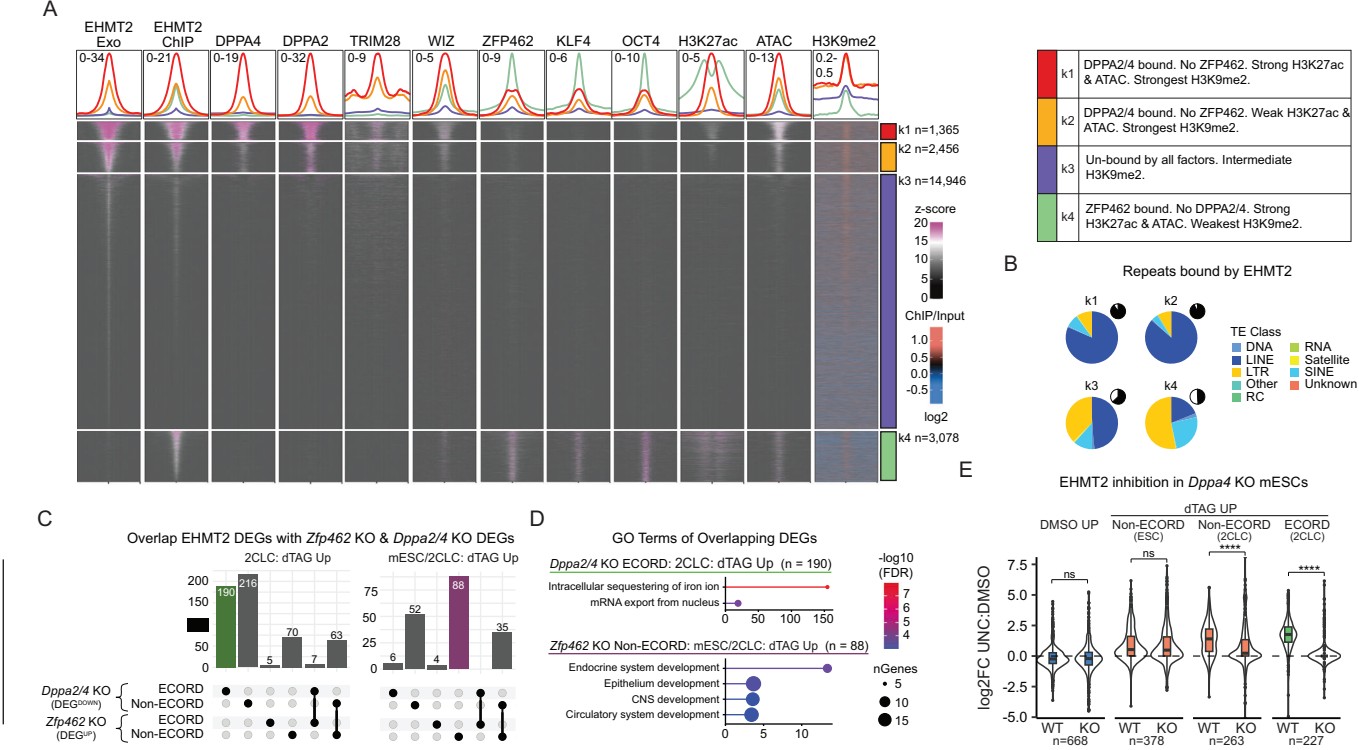

**Figure 5. EHMT2 binding suggests locus-specific modes of gene regulation.**

(A) k-means clustering of EHMT2 ChIP-Exo and ChIP-seq signal in bulk mESCs with putative co-factors and factors of interest (for a list of datasets used, see Dataset EV6). Clustering was performed on a union set of ChIP-Exo and ChIP peaks. Heatmaps are +/− 2 kb from the EHMT2 peak summit, except for H3K9me2, which is +/− 50 kb. For peaks called in both Exo and ChIP, the midpoint of the summits was used instead. Z-score values apply to all columns except for H3K9me2, which is represented as ChIP/input. A summary of the properties of EHMT2 k-means clusters is included on the right. (B) Classes of repetitive elements (TEs) that EHMT2 binds. Pie-chart insets indicate the fraction of each peak category that overlaps a repeat. (C) Overlap of DEGs with genes downregulated in mESCs upon KO of either *Dppa2* or *Dppa4* (GSE126920) (De Iaco et al, 2019) or upregulated upon *Zfp462* KO (GSE176321) (Yelagandula et al, 2023). The UpSet plots, located below the bar graphs, define the gene categories represented in each column. (D) GO Terms of DEG categories highlighted in (D). (E) RNA-seq log2 Fold Changes (log2FC) of 7 days EHMT2-dTAG DEGs in *Dppa4* KO or parental J1 (WT) mESCs after culture in the presence of the EHMT2 inhibitor UNC0638 (UNC) or DMSO. Gene categories (ECORD/Non-ECORD, mESC, and 2CLC) were defined by their behavior in the 2CLC/mESC RNA-seq analysis (see Fig. 3). The numbers of genes in each category are indicated. ****$P < 0.0001$ with one-way ANOVA. Exact $P$ values are listed in Dataset EV6. Boxplots are defined in Fig. 2. Source data are available online for this figure.

candidate chromatin-associated trans-acting factors (DPPA2/4, ZFP462, TRIM28, WIZ), pluripotency-associated TFs (OCT4, KLF4), and chromatin features that mark active regions of the genome (H3K27ac, ATAC-seq). We also integrated our H3K9me2 ULI-NChIP data from bulk mESC cultures.

K-means clustering (Fig. 5A; Dataset EV4) revealed four distinct categories of EHMT2 peaks predominantly defined by the mutually exclusive presence of either the zinc finger TF ZFP462, previously suggested to direct EHMT2 in mESCs towards germ layer-associated genes for repression (Yelagandula et al, 2023), and the heterodimeric TFs DPPA2/4, previously shown to be required for the activation of ZGA-associated transcripts in mESCs (De Iaco et al, 2019). Thus, K1 EHMT2 peaks were detected by both ChIP-seq and ChIP-exo, exhibiting strong binding of DPPA2/4, TRIM28, and WIZ, as well as strong H3K27ac, ATAC-seq signals, and a strong H3K9me2 enrichment around the peak and the surrounding genomic window. K2 was similar to K1 with globally lower occupancy levels of all factors and weaker H3K27ac and ATAC-seq signals but comparable H3K9me2 levels. More than 85% of K1 and K2 sites localized to H3K9me2 domains (Appendix Fig. S5A) and occurred almost exclusively at TEs with a striking enrichment for

LINE elements (Fig. 5B). K3 peaks were primarily detected by ChIP-exo and showed no association with any of the tested TFs (Fig. 5A) but were robustly marked by H3K9me2 and preferentially localized to H3K9me2 domains (Appendix Fig. S5A); K4 peaks were predominantly detected by ChIP-seq, featured binding of ZFP462 and pluripotency-associated factors, and exhibited strong H3K27ac and ATAC-seq signals in a pattern characteristic of active enhancers and promoters (Fig. 5A). K4 sites were enriched for LTRs but depleted for LINEs (Fig. 5B). Although K4 sites exhibited a local enrichment of H3K9me2 signal around the EHMT2 peak summit (Fig. 5A), they showed only weak overlap with H3K9me2 domains relative to the genome (Appendix Fig. S5A). ChromHMM analysis revealed heterochromatic features at K1/K2 and euchromatic features at K4, with K3 occupying an intermediate state (Appendix Fig. S5B). Notably, weak enhancer features were observed across all peak categories (Appendix Fig. S5B), possibly suggesting that context-dependent activation ability is a commonality among EHMT2-bound sites. Additionally, the plurality of binding sites in all four clusters was localized to gene bodies and intergenic sites (Appendix Fig. S5C), consistent with a predominantly promoter-distal gene-regulatory function of EHMT2.

Overall, our k-means clustering identifies distinct categories of EHMT2 binding sites with unique chromatin features, suggesting they may have different gene-regulatory functions.

To determine how genes transcriptionally affected by EHMT2 depletion associate with distinct EHMT2 binding modes, we integrated our ChIP categories with our RNA-seq DEGs (as defined in Fig. 3I). This revealed a pronounced over-representation of K1/K2 (bound by DPPA2/4s) and K3 (bound by none of the tested co-factors) peaks at 2CLC$^{dTAG\_UP}$ DEGs, which was evident at gene loci both inside and outside of ECORDs (Appendix Fig. S5D,E). In contrast, K4 peaks (bound by ZFP462) were overrepresented around mESC/2CLC$^{dTAG\_UP}$ genes outside ECORDs (Appendix Fig. S5D,E). This suggested potentially distinct functions of DPPA2/4 and ZFP462 in regulating specific subsets of EHMT2 target loci. In line with this observation, many 2CLC$^{dTAG\_UP}$ DEGs were downregulated in DPPA2/4 KO mESCs but remained largely unaffected by KO of ZFP462 (Fig. 5C). This difference was particularly pronounced for 2CLC$^{dTAG\_UP}$ genes within ECORDs (Fig. 5C). In contrast, mESC/2CLC$^{dTAG\_UP}$ DEGs, which predominantly localize outside of ECORDs, were more sensitive to the loss of ZFP462 and experienced upregulation in mESCs lacking this TF (Fig. 5C). A small fraction of these genes was downregulated in DPPA2/4 KO mESCs (Fig. 5C), suggesting a higher degree of regulatory heterogeneity for mESC/2CLC$^{dTAG\_UP}$ DEGs than for ECORDs. Functionally, 2CLC$^{dTAG\_UP}$ DEGs downregulated in DPPA2/4 KO were enriched for GO Terms related to iron and mRNA metabolism (Fig. 5D; Dataset EV5). At the same time, mESC/2CLC$^{dTAG\_UP}$ DEGs upregulated upon ZFP462 KO were enriched for GO terms related to organogenesis, such as nervous and circulatory system development (Fig. 5D; Dataset EV5). These observations suggest distinct biological functions of ZFP462-controlled and DPPA2/4-controlled transcriptional programs.

We further investigated the requirement of DPPA2/4 for the activation of EHMT2-repressed genes. To do so, we exposed DPPA4 KO and parental J1 mESCs to UNC0638, as our prior results had shown that the enzymatic activity of EHMT2 is required for ECORD repression and to counteract the mESC-to-2CLC transition (Fig. 4E; Appendix Fig. S4J). RNA-seq analysis revealed that mESC-associated EHMT2 targets were still upregulated upon EHMT2 inhibition in DPPA4 KO mESCs (Fig. 5E). In contrast, these cells failed to upregulate 2CLC-associated EHMT2 targets inside and outside of ECORDs (Fig. 5E), even though high sample-to-sample variability limited the reliable identification of DEGs (Dataset EV2). These observations demonstrate that the loss of EHMT2 is insufficient for the derepression of ECORDs and other 2CLC-associated genes in the absence of DPPA4.

Together, our analyses suggest that EHMT2 binding in ESCs represses two broadly distinct categories of genes: (1) genes normally active later during development are repressed by ZFP462 acting as a co-factor for EHMT2, and (2) genes active during ZGA whose expression in 2CLCs requires DPPA2/4 and is antagonized by EHMT2. A significant fraction of the latter is organized in ECORDs.

## EHMT2 regulates the 2-cell-like state, in part, by antagonizing DPPA2/4 binding and activity

Our analysis has shown that antagonism between EHMT2-mediated repression and DPPA2/4-mediated activation is involved in determining the expression status of ECORDs and, thus, the rate of the mESC-to-2CLC transition. Since *Dppa4* is expressed in both mESCs and 2CLCs (Appendix Fig. S6A; Dataset EV4), we hypothesized that EHMT2 in mESCs might antagonize DPPA2/4 binding or activity within ECORDs. To test this model, we performed CUT&RUN (Skene and Henikoff, 2017) for the endogenous DPPA4 protein in purified mESCs (MERVL−) and 2CLCs (MERVL+)(Appendix Fig. S6B). We did that in the context of EHMT2 depletion (dTAG-13 treatment) to obtain sufficient numbers of 2CLCs for analysis. This revealed that DPPA4 bound extensively throughout the genome (a total of 42,648 sites) and exhibited high dynamicity between mESCs and 2CLCs, with more than 50% of binding sites being specific to either stage (Appendix Fig. S6C; Dataset EV4). Consistent with our observations with other TFs (Di Giammartino et al, 2019), binding sites common to both cell types were enriched for promoters, while developmental stage-specific sites tended to overlap introns or intergenic sites (Appendix Fig. S6D). Compared to the genome-wide binding of DPPA4, which occurred primarily at sites either specific to mESCs or shared between mESCs and 2CLCs (Appendix Fig. S6C), DPPA4 binding sites at ECORDs were more frequently 2CLC-specific (Fig. 6A). This suggests that the derepression of ECORDs during the transition from mESCs to 2CLCs may be regulated in part by the redistribution of DPPA4 to newly accessible sites.

What are 2CLC-specific DPPA4 binding sites? 2CLC-specific DPPA4 binding near ECORDs frequently overlapped TEs (Fig. 6A) with enrichment for LTRs (Fig. 6B; Appendix Fig. S6E) and in particular MERVL elements (Appendix Fig. S6F). MERVL elements are active in 2CLCs and have been suggested to act as alternative transcription start sites or enhancers (Zhang et al, 2019) that can increase the expression of nearby genes (Fort et al, 2014; Macfarlan et al, 2012). Consistent with a positive gene-regulatory function in 2CLCs, 2CLC-specific DPPA4-bound MERVLs were located more proximally to ECORD DEGs when compared to common or mESC-specific sites (Fig. 6C,D).

We hypothesized that EHMT2 in mESCs might directly antagonize DPPA4 binding by occupying its binding sites and displacing DPPA4, a notion supported by the reported role of EHMT2 in suppressing MERVL expression (Maksakova et al, 2013). In contrast to these expectations, though, very few (<5%) 2CLC-specific DPPA4 sites were occupied by EHMT2 in mESCs, which argues against our hypothesis of EHMT2 directly acting on these sites (Fig. 6A; Appendix Fig. S6C). To establish the relationship between EHMT2, DPPA4, and ECORD activation, we next used ULI-NCHIP to assay H3K9me2 in purified mESCs and 2CLCs (Appendix Fig. S7A), which revealed globally similar levels of H3K9me2 in both cell types that were reduced by about 40–50% by EHMT2 depletion (Appendix Fig. S7B,C). In these experiments, we utilized the RAR agonist TTNBP (TT) to obtain sufficient numbers of viable 2CLCs (see also Appendix Fig. S3D). Importantly, analysis by flow cytometry revealed changes in global H3K9me2 levels between mESCs and 2CLCs in the absence or presence of TT (Appendix Fig. S7D). At the level of EHMT2 targets, we observed the highest H3K9me2 levels in mESCs at 2CLC-associated ECORDs (Fig. 7A; Appendix Fig. S7E), followed by other 2CLC-associated genes and mESC-associated loci repressed by EHMT2 (Fig. 7A). Despite the lack of direct EHMT2 binding, 2CLC-specific DPPA4 binding sites were strongly marked by H3K9me2 (Fig. 7B; Appendix Fig. S7F) and localized to H3K9me2 domains (Fig. 7C). This indicates that distal EHMT2 binding may initiate the formation of H3K9me2 domains that

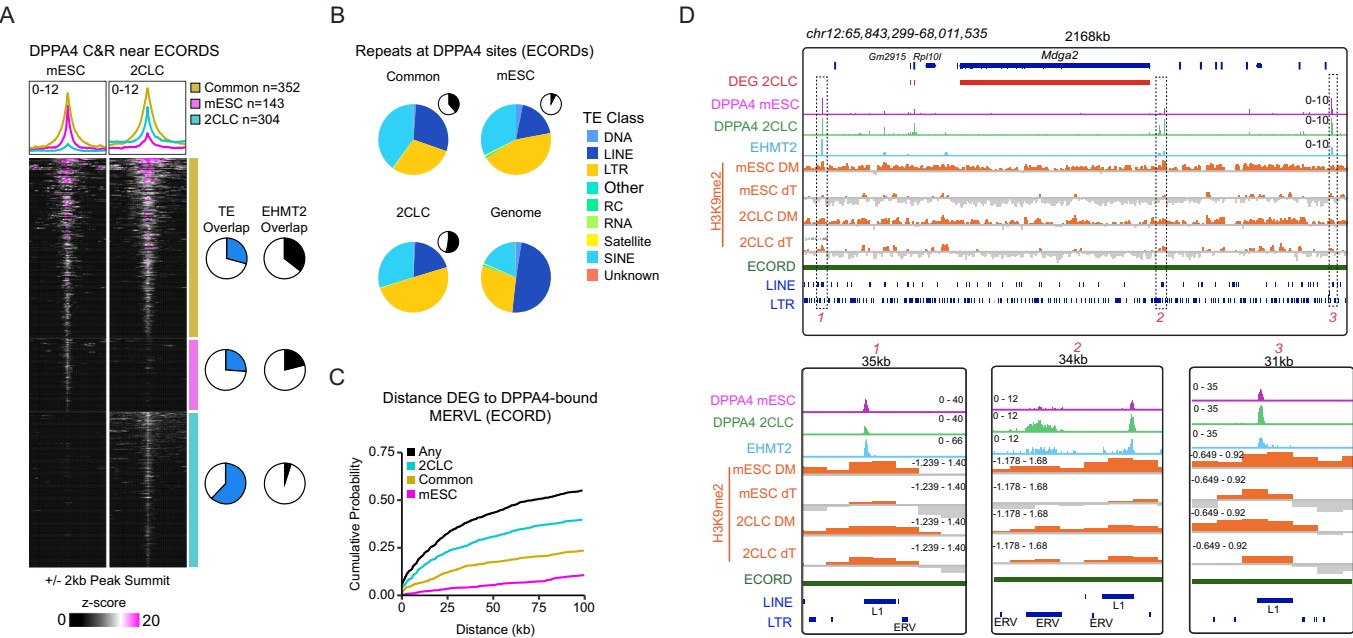

**Figure 6. EHMT2 antagonizes DPPA2/4 function at ECORDs.**

(A) DPPA4 C&R signal ($+/-$ 2kb of peak summits) around peaks associated with ECORDs. ECORD-associated peaks were identified by assigning DPPA4 peaks to the nearest expressed gene. For common peaks, the mid-point of the summits was used. (B) Classes of repetitive elements (TEs) that DPPA4 binds. Pie chart insets indicate the fraction of all peaks that each category represents. Boxplots are defined in Fig. 2. (C) Distance of ECORD genes to the nearest DPPA4-bound MERVL element. For this analysis, all DPPA4 peaks were used. MERVL: "ERVL" and "ERVL-MaLR" repeat families. (D) Browser shots of EHMT2 ChIP-Exo, DPPA4 C&R, and H3K9me2 ChIP (DM-DMSO, dT-dTAG treated (top). Bottom: Zoomed-in views of highlighted regions 1, 2, and 3 (dashed boxes).

spread through ECORDs, thereby occluding DPPA4 binding sites at MERVLs proximal to 2CLC genes. We observed that EHMT2 binding sites nearest to DPPA4-bound MERVLs overlapped LINE-1 elements (Appendix Fig. S7G) and that EHMT2-bound LINE-1 in mESCs were characterized by significantly elevated H3K9me2 levels that extended from the EHMT2 peak into their genomic vicinity (Fig. 7D). These LINE-1 sites bound by EHMT2 localized in more gene distal positions than 2CLC-specific DPPA4 sites (Fig. 6D) and comprised predominantly full-length L1Md_T/A subtypes (Appendix Fig. S7H). At ECORDs, EHMT2-bound L1s—but not other sites bound by EHMT2—were strongly enriched near gene-proximal MERVLs bound by DPPA4 in 2CLCs. In contrast, we found no enrichment for EHMT2-bound L1s at 2CLC-specific DPPA4 sites near DEGs outside of ECORDs (Appendix Fig. S7G). Many of these L1s were also bound by DPPA4 (Fig. 6D). These observations suggest that EHMT2-bound L1s play a regulatory role during the silencing of ECORDs in mESCs and their DPPA4-mediated reactivation during the mESC-to-2CLC transition.

Of note, 2CLC-associated EHMT2 targets (Fig. 7; Appendix Fig. S7E) and MERVLs (Fig. 7B) only exhibited moderately reduced levels of H3K9me2 in spontaneously arising 2CLCs compared to mESCs. Accordingly, the enzymes regulating H3K9me2 levels (including *Ehmt2*) are expressed at similar levels in mESCs and 2CLCs (Appendix Fig. S6A). This is in stark contrast to mouse embryos at the 2-cell stage, which express significantly lower levels of *Ehmt2* than mESCs (Appendix Fig. S6A) and have been shown to exhibit a low abundance of H3K9me2 (Deng et al, 2020; Zylicz et al, 2018). Nevertheless, EHMT2-bound L1s experienced a pronounced drop of H3K9me2 levels in 2CLCs centered around the site of

EHMT2 binding (Fig. 7D). EHMT2 depletion resulted in no further reduction of H3K9me2 at these sites in 2CLCs. At the same time, regions further away from the EHMT2 peak showed progressively reduced methylation in the absence of EHMT2 (Fig. 7D). These observations suggest that the mESC-to-2CLC transition might be associated with the exclusion of EHMT2 activity from L1 elements in ECORDs or with a local remodeling of H3K9me2-marked histones. In addition, other histone methyltransferases might contribute to the maintenance of H3K9 methylation at L1s in ECORDs in the absence of EHMT2, as suggested by the binding of TRIM28 to these elements (Fig. 5A).

In summary, we propose two distinct binding behaviors and repressive modalities of EHMT2 in mESCs. The binding of EHMT2 to gene distal LINE-1 elements in mESCs nucleates broad H3K9me2 domains that extend over clusters of 2CLC genes (ECORDs) and counteract the binding of DPPA2/4 and possibly other activating factors to proximal, LTR-derived gene-regulatory elements. These H3K9me2 domains are weakened during the spontaneous mESC-to-2CLC transition and fully resolved upon EHMT2 depletion (Fig. 7E). In contrast, EHMT2 also represses a subset of germ layer-associated genes in mESCs through ZFP462-mediated binding to candidate enhancer elements and deposition of local H3K9me2 (Appendix Fig. S7I).

# Discussion

Unlike TFs, most chromatin-modifying proteins lack DNA sequence specificity and therefore have target genes and functions

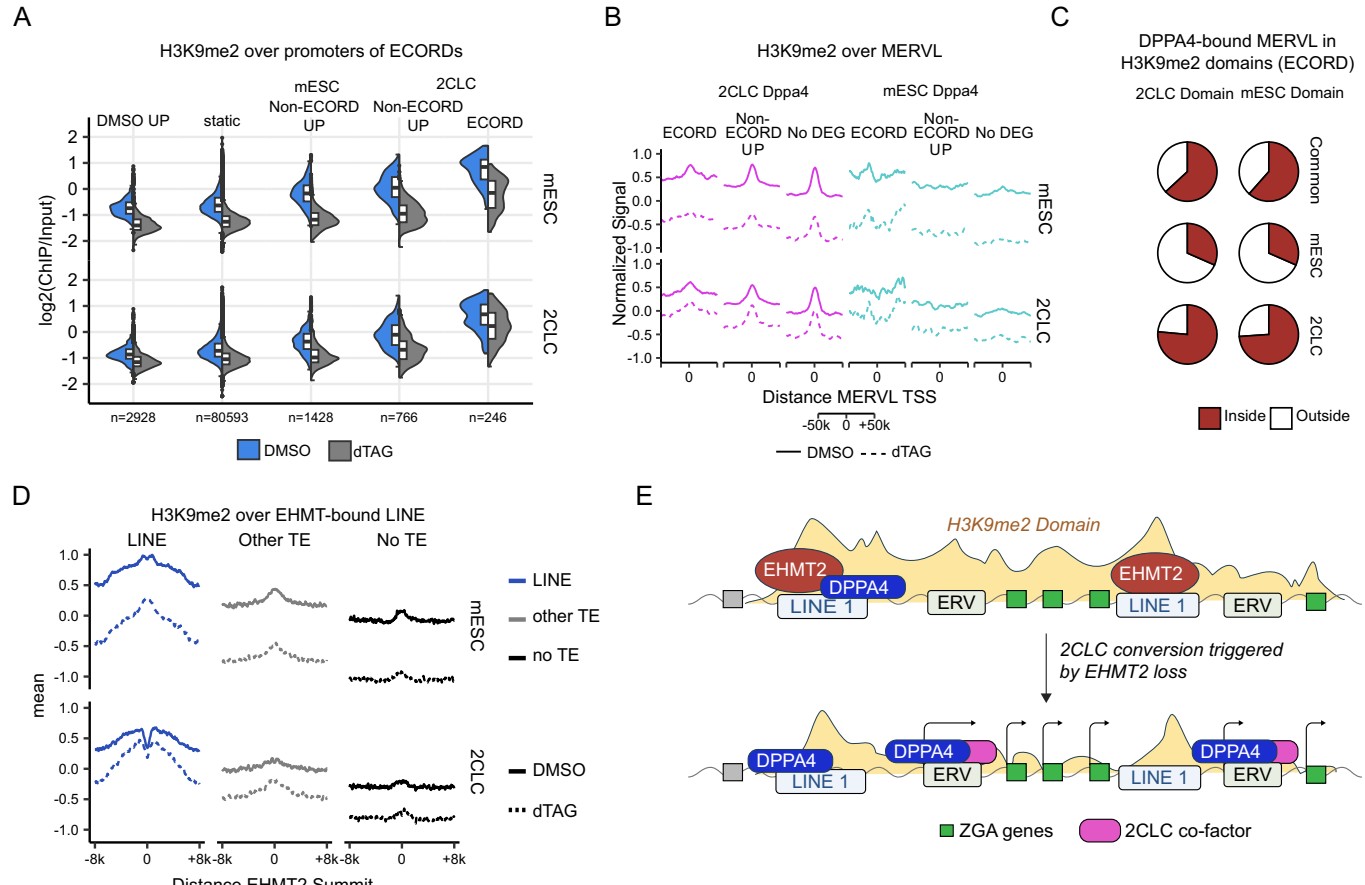

**Figure 7. Evidence for a regulatory role of EHMT2-bound L1s for ECORD expression.**

(**A**) H3K9me2 signal in mESCs and 2CLCs (DMSO, dTAG) over gene promoters (TSS) of 7 days dTAG genes. Gene categories (ECORD/Non-ECORD, mESC, and 2CLC) were defined by their behavior in 2CLC/mESC RNAseq, as in Fig. 3I. Three biological replicates were used. The TSSs of all expressed genes (n = 86,064) were used and categorized as indicated. (**B**) Average signal plots of H3K9me2 signal around the promoters (TSS) of DPPA4-bound MERVLs in DMSO- (solid line) or dTAG- (dotted line) treated mESCs (top) and 2CLCs (bottom), distinguishing between MERVLs bound only in 2CLCs (purple) or only in mESCs (cyan). MERVLs were assigned to genes using H3K9me2 domains. (**C**) Proportion of DPPA4-bound MERVL peaks that are in H3K9me2 domains. (**D**) Average signal plots of H3K9me2 signal in mESCs and 2CLC (DMSO, dTAG) around EHMT2 peak summits, split by whether the EHMT2 peak overlaps an L1 LINE, a different type of repeat (other TE), or no repeat (no TE). (**E**) Model of EHMT2 regulation of ECORDs. For all plots, the H3K9me2 signal is ChIP/Input-normalized and then multiplied by an additional scaling factor to account for global differences in H3K9me2 levels (see "Methods").

that vary considerably across cell types. We have combined acute protein depletion with different genomics assays to dissect the role of the chromatin repressor EHMT2 in naive mouse pluripotent stem cells. Our findings suggest that EHMT2 restricts the bidirectional differentiation capacity of mESCs, counteracting both the activation of gene loci associated with "forward" germ layer differentiation and those highly expressed in 2CLCs, which are related to early post-fertilization development and ZGA. While the significantly increased rate of mESC-to-2CLC transitions upon EHMT2 loss underscores the relevance of repressing the latter group of targets, we observed no overt germ-layer differentiation in EHMT2-depleted mESCs. This may suggest that EHMT2 in mESCs represses specific gene loci associated with different lineages, rather than comprehensive gene expression programs. Alternatively, our observations align with redundant mechanisms of gene repression at developmental gene loci, as suggested by studies into the interaction of EHMT2 and PRC2 in mESCs (Mozzetta et al, 2014).

In addition to their broadly divergent biological functions, the two groups of EHMT2 target genes we characterized differ in their genomic organization and the cis-regulatory elements and co-factors with which EHMT2 engages to modulate their expression. About 30% of ZGA-associated transcripts upregulated upon EHMT2 depletion are organized into what we coined ECORDs—clusters of co-regulated genes in heterochromatic regions of the genome that are poor in genes and rich in TEs. EHMT2-catalyzed H3K9me2 has been chiefly associated with gene silencing in euchromatin regions of the genome (Fukuda et al, 2021; Peters et al, 2003; Rice et al, 2003). The existence of ECORDs suggests that EHMT2 also directly regulates developmentally controlled non-euchromatic regions in mESCs, consistent with global changes in H3K9me2 levels in EHMT2 KO mESCs that are not observed in differentiated cells (Yan et al, 2020). ECORDs might, therefore, represent a functionally and regulatory distinct state of heterochromatin (Spracklin et al, 2023).

The activation of ECORDs in mESCs is intimately linked to the acquisition of 2CLC identity—about a third of transcripts that distinguish 2CLCs from mESCs are organized in clusters of varying size and complexity. While the activation of specific gene clusters, such as ZSCAN4 (Falco et al, 2007) and OBOX (Ji et al, 2023), has been strongly associated with the 2CLC stage, our findings establish clustered gene expression as a prevalent feature of this cell stage. Furthermore, the elevated expression of genes encoded by ECORDs during ZGA suggests that these gene clusters represent regulatory units that characterize the 2-cell state in the mouse embryo. Functionally, the organization of genes into ECORDs may enable their rapid activation and coordinated shutdown during development (Vega-Sendino et al, 2024). In this manner, ECORDs may be loosely analogous to Hox gene clusters, as they represent clusters of genes with shared functions that are coordinately deactivated through the establishment and spreading of heterochromatin domains (Hubert and Wellik, 2023). Several ECORD-encoded genes, such as ZSCAN4 (Zhang et al, 2019) and OBOX4 (Ji et al, 2023), have regulatory functions in 2CLCs and affect early development when inactivated. This supports the importance of tight developmental control over ECORDs and raises the possibility that their study could reveal novel 2CLC/ZGA regulators.

EHMT2 contributes to the control of ECORD expression in an antagonistic relationship with DPPA2/4, a heterodimeric pair of TFs required for the expression of ZGA-associated genes in mESCs (De Iaco et al, 2019). Evolutionarily young LINE-1 elements within ECORDs may be central to this antagonism, as they bind both DPPA2/4 and EHMT2 and undergo local H3K9me2 remodeling during the mESC-to-2CLC transition. In addition, our ChIP and CUT&RUN analyses support the notion that EHMT2 in mESCs prevents DPPA2/4 binding to MERVL-derived promoters or promoter-proximal gene-regulatory elements within the context of broad H3K9me2 domains. DPPA2/4 binding to these promoters or their activation likely requires additional factors, such as DUX, which is involved in the upregulation of ZGA-associated genes during the mESC-to-2CLC transition (Hendrickson et al, 2017; Yang et al, 2020).

Since we did not observe EHMT2 binding to DPPA4-bound MERVLs in mESCs, it seems likely that H3K9me2 spreads from distal sites, such as local LINE-1 elements that are strongly marked by H3K9me2 and bound by EHMT2. However, we cannot rule out that our analysis missed sporadic EHMT2 binding to MERVLs. Notably, many EHMT2-bound LINE-1 elements are also bound by other repressors, such as TRIM28 (Rowe et al, 2010), and by activators, such as DPPA2/4, suggesting that these elements serve as docking sites for transcriptional regulators with diverse, and in some instances opposing, functions. We found that spontaneously emerging 2CLCs still harbor substantial amounts of H3K9me2 genome-wide and within ECORDs, an apparent molecular difference from 2-cell-stage mouse embryos (Deng et al, 2020; Zylicz et al, 2018). This may indicate the existence of multiple, partially redundant mechanisms of gene repression, including other histone methyltransferases, as well as molecular mechanisms of gene activation capable of overriding H3K9me2 function, such as the detachment of heterochromatic regions from the nuclear lamina (Marin et al, 2025). The presence of additional barriers to the 2CLC state is also consistent with the observation that EHMT2 loss alone converts only a minority of mESCs.

Control of the activation status of LINE-1 elements has been suggested to be essential for successful early mouse preimplantation development (Jachowicz et al, 2017). In addition, LINE-1s have

been recently proposed to serve as distal enhancers that control the expression of ZGA-associated genes such as *Zscan4* (Li et al, 2024). This is consistent with the low level of H3K27ac and chromatin accessibility we have observed at EHMT2-bound k1 and k2 sites in bulk mESC cultures. Therefore, in addition to using LINE-1s as nucleation sites for H3K9me2 spreading, EHMT2 might counteract ECORD activation by interfering with the latent enhancer activity of these elements. In accordance with such a function, LINE-1s within ECORDs experience local loss of H3K9me2 at EHMT2 binding sites during the mESC-to-2CLC conversion. Deletion experiments or the targeted recruitment of specific chromatin-modifying activities in 2CLCs and mESCs should help to clarify the stage-specific molecular functions of these elements.

Outside of ECORDs, EHMT2 depletion resembles aspects of the transcriptional consequences seen in ZFP462 KO mESCs. This aligns with a model of gene regulation in which ZFP462 recruits EHMT2 to repress germ-layer-associated target genes, as recently proposed for mesendodermal genes in mESCs (Yelagandula et al, 2023). Our observations broadly confirm these prior findings and also demonstrate that EHMT2 represses neuroectodermal gene loci in mESCs, likely via ZFP462. This is relevant in the context of the established role of EHMT1, the heterodimeric partner of EHMT2, in neurodevelopmental disorders (Kleefstra et al, 2006). We do not rule out the existence of additional EHMT2-recruiting factors required to silence germ-layer-associated genes in mESCs, such as REST (Mozzetta et al, 2014; Roopra et al, 2004).

In summary, our experiments have provided molecular insight into the role of the chromatin repressor EHMT2 in counteracting 2CLC-specific and lineage-associated gene expression. It remains to be determined whether the distinct modes of EHMT2 target gene binding and repression identified in our study are also operative in other contexts, such as neurodevelopment, immune cell metabolism, and various cancers, where EHMT2 plays critical cellular functions. The identification of ECORDs, which encode many poorly characterized transcripts, also offers a novel opportunity to reveal gene-regulatory aspects of ZGA in a tractable experimental system.

# Methods

## Reagents and tools table

| Reagent/resource | Reference or source | Identifier or catalog number |
|---|---|---|
| **Cell lines** | | |
| Mouse: EHMT2-dTAG mESCs | This manuscript | n.a. |
| Mouse: MERVL-GFP mESCs | This manuscript | n.a. |
| Mouse: DPPA4 KO mESCs | Didier Trono, EPFL, Lausanne | n.a. |
| Mouse: J1 mESCs | Didier Trono, EPFL, Lausanne | n.a. |
| **Antibodies** | | |
| HA | Cell Signaling Technology; Abcam | CST 3724S (RRID:AB_1549585); ab9110 (RRID:AB_307019) |
| H3K9me2 | Abcam | ab1220 (RRID:AB_449854) |

| Reagent/resource | Reference or source | Identifier or catalog number |
|---|---|---|
| SSEA1, Biotin-conjugated | Thermo Fisher | 13-8813-82 (RRID:AB_657625) |
| EpCAM (CD326), PECy7-conjugated | Thermo Fisher | 25-5791-80 (RRID:AB_1724047) |
| ZSCAN | Millipore-Sigma | AB4340 (RRID:AB_2827621) |
| DPPA4 | Novus | AF3730 (RRID:AB_2094166) |
| Donkey anti-Rabbit Alexa Fluor 555 | Thermo Fisher Scientific | A-31572 (RRID:AB_162543) |
| Donkey anti-Mouse Alexa Fluor 647 | Thermo Fisher Scientific | A-31571 (RRID:AB_162542) |
| **Chemicals** | | |
| DSG (Di(N-succinimidyl) glutarate | Sigma | 80424-5MG-F |
| dTAG13 | Tocris | 6605 |
| DMSO | Sigma | D8418 |
| UNC0638 | Selleckchem | S8071 |
| Pladienolide B | Tocris | 6070 |
| 1-azakenpaullone | Selleck | S7193 |
| WS6 | Selleck | S7442 |
| TTNPB | Selleck | S4627 |
| **Commercial assays** | | |
| eBioscience™ Foxp3/ Transcription Factor Staining Buffer Set | Thermo Fisher Scientific | 00-5523-00 |
| KAPA hyper prep-kit | Roche | KK8502 |
| **Software and algorithms** | | |
| snakemake | Mölder et al, 2021 | https://anaconda.org/ bioconda/snakemake |
| Trim Galore | Github | https://github.com/ FelixKrueger/TrimGalore |
| STAR | Dobin et al, 2013a | https://anaconda.org/ bioconda/star |
| samtools | Li et al, 2009 | https://anaconda.org/ bioconda/samtools |
| Picard | Broad Institute, 2018 | https://anaconda.org/ bioconda/picard |
| bedtools | Quinlan and Hall, 2010 | https://anaconda.org/ bioconda/bedtools |
| BBTools | BBTools | https://anaconda.org/ bioconda/bbmap |
| Deeptools | Ramirez et al, 2014 | https://anaconda.org/ bioconda/deeptools |
| TE Transcripts | Jin et al, 2015a | https://anaconda.org/ bioconda/tetranscripts |
| MACS2 | Zhang et al, 2008 | https://anaconda.org/ bioconda/macs2 |
| org.Mm.eg.db | Carlson, 2021 | https://bioconductor.org/ packages/release/data/ annotation/html/ org.Mm.eg.db.html |
| rtracklayer | Lawrence Gentleman and Carey, 2009 | https://bioconductor.org/ packages/release/bioc/html/ rtracklayer.html |

| Reagent/resource | Reference or source | Identifier or catalog number |
|---|---|---|
| GenomicRanges | Lawrence et al, 2013 | https://bioconductor.org/ packages/release/bioc/html/ GenomicRanges.html |
| GenomicFeatures | Lawrence et al, 2013 | https://bioconductor.org/ packages/release/bioc/html/ GenomicFeatures.html |
| plyranges | Lee Cook and Lawrence, 2019 | https://bioconductor.org/ packages/release/bioc/html/ plyranges.html |
| seqplots | Stempor and Ahringer, 2016 | https://github.com/ cmuyehara/seqplots |
| DESeq2 | Love Huber and Anders, 2014 | https://bioconductor.org/ packages/release/bioc/html/ DESeq2.html |
| ChIPpeakAnno | Zhu et al, 2010 | https://bioconductor.org/ packages/release/bioc/html/ ChIPpeakAnno.html |
| kmed | Budiaji, 2022 | https://cran.r-project.org/ web/packages/kmed/ index.html |
| tidyverse | Wickham et al, 2019 | https://cran.r-project.org/ web/packages/tidyverse/ index.html |
| ggplot2 | Wickham, 2010 | https://cran.r-project.org/ web/packages/ggplot2/ index.html |
| patchwork | Pedersen, 2024 | https://cran.r-project.org/ web/packages/patchwork/ index.html |
| eulerr | Larsson, 2024 | https://cran.r-project.org/ web/packages/eulerr/ index.html |
| ComplexUpset | Krassowski, 2020 | https://cran.r-project.org/ web/packages/ ComplexUpset/index.html |
| fishualize | Schiettekatte et al, 2019 | https://cran.r-project.org/ web/packages/fishualize/ index.html |
| ggalluvial | Brunson and Read, 2020 | https://cran.r-project.org/ web/packages/ggalluvial/ index.html |
| rstatix | Kassambara, 2023 | https://cran.r-project.org/ web/packages/rstatix/ index.html |
| ImageJ | Schneider Rasband and Eliceiri, 2012 | https://imagej.net/ij/ |
| rrvgo | Sayols, 2023 | https:// www.bioconductor.org/ packages/release/bioc/html/ rrvgo.html |

## Mouse cell lines

The parental mouse ESC lines used for gene targeting were KH2 (Beard et al, 2006) on a C57BL/6J x 129S1 F1 background. DPPA2KO, DPPA4KO, and WTJ1 cells were kindly provided by the Trono Lab (De Iaco et al, 2019). All cell lines were

authenticated by PCR genotyping and routinely tested for mycoplasma.

## Mouse ESC culture

ESCs were cultured in KO DMEM (Gibco 10829018) supplemented with 15% FBS (Gemini Benchmark), 2 mM Glutamax (Gibco 35050079), 0.1 mM nonessential amino acids (Gibco 11140076), 100 mg/ml penicillin/streptomycin (Gibco 15140163), 0.1 mM 2-mercaptoethanol (Gibco 21985023), and 1000U/ml leukemia inhibitory factor prepared in-house. Cells were cultured on a feeder layer of mitomycin C-treated mouse embryonic fibroblasts (MEFs) on gelatin-coated plates. In depletion experiments, cells were cultured in the presence of dTAG-13 (200 nM) for the indicated periods. For chemical inhibition of EHMT2, cells were cultured in mESC media supplemented with UNC0638 (1 mM). Cells cultured in DMSO served as controls. To modulate additional pathways associated with the mESC-to-2CLC conversion, mESCs were cultured in the presence of Pladienolide B (PB; 2.5 nM), 1-azakenpaullone (1-AP; 2.5 mM), WS6 (0.5 mM), or TTNBP (TT; 0.2 M) for 48 h before analysis.

## Generation of EHMT2-dTAG mESCs

To generate EHMT2-dTAG mESCs, homology arms covering about 1.2 kb of sequence around the *Ehmt2* stop codon were PCR-amplified from KH2 genomic DNA and cloned into pBluescript (Stratagene) vector together with an FKBP12F36V-2xHA-P2A-NLS-mCherry cassette, using Gibson assembly. Parental KH2 mESCs were co-transfected with the targeting vector and a pX330-neo$^R$ vector expressing Cas9 and gRNAs targeting *Ehmt2* using TransIT-293 (Mirus Bio 2700). The next day, cells were plated at low density on a 10 cm plate and cultured in selection media containing Geneticin (500 µg/ml) for 48 h. Individual clones were picked, expanded, and confirmed using PCR, Sanger sequencing of PCR amplicons, and flow cytometry. Guide RNAs are listed in Dataset EV6.

## Generation of MERVL-GFP mESCs

EHMT2-dTAG mESCs were transfected with the 2C-3XtbGFP-PEST plasmid (Ishiuchi et al, 2015) (Addgene #69072) using lipofection with TransIT-293 (Mirus Bio 2700). Cells were seeded at a clonal density and selected in G418 (250 µg/ml) for 5 days before the cultures were inspected under an EVOS fluorescence microscope. Individual colonies with rare GFP$^+$ cells were then picked for further expansion.

## Western blotting

Three independent clones were treated with DMSO or dTAG for 24 h before protein isolation. Cells were washed with PBS−/− and harvested using Trypsin (Life Technologies 25200114). Nuclear lysates were prepared using the NE-PER kit (Thermo Fisher 78835) according to the manufacturer's instructions. Histones were isolated using the Histone Extraction Kit (Abcam ab113476) according to the manufacturer's instructions. Protein concentration was measured using the Bradford Reagent (BioRad 5000201). Samples were boiled in Laemmli Sample Buffer (BioRad) with beta-mercaptoethanol and then run on

Invitrogen precast gels. Blots were imaged using Azure Biosystems C400. Images were quantified using ImageJ (Schneider Rasband and Eliceiri, 2012). The following antibodies were used at a dilution of 1:1000: anti-HA (Abcam 9110), anti-H3K9me2 (ab1220), and anti-histone H3 (Abcam 1791).

## Flow cytometry

Expression of mCherry in EHMT2-dTAG mESCs was determined using the 561 nm (610/20) channel on a BD Fortessa. For the EHMT2 and H3K9me2 depletion kinetics cells, two independent clones were treated with DMSO or dTAG (200 nM) for the indicated periods. 500k cells from each treatment group (technical triplicates) were collected in a single-cell suspension using a Thermo Fisher intracellular staining kit (cat. 00-5523-00). Cells were fixed for 25 min and incubated with 50 µL of primary antibodies diluted in permeabilization buffer for HA (CST-3724S, 1:800) and H3K9me2 (ab1220, 1:400) for 30 min. Cells were then incubated with 50 µL secondary antibodies at a 1:500 dilution (A-31572 555aRb, A-31571 647aMs) for 30 min in the dark. Samples were run through the BD Fortessa flow cytometer and analyzed using the FlowJo software. Significance was called using the R t_test default function in the rstatix package (Kassambara, 2023). For the EHMT2-HA washout recovery flow, cells were treated with DMSO or dTAG in duplicates for 24 h and then replaced with media without treatment for the indicated time periods. Samples were processed and analyzed as above. Cells were stained with HA (CST-3724S 1:800) primary and A-21206 488aRb secondary antibodies. The secondary only control was incubated with the secondary antibody. For Zscan4 and H3K9me2 analysis, EHMT2-dTAG mESCs were treated with DMSO or dTAG in triplicate for 48 h and processed as above. Cells were stained with H3K9me2 (ab1220) and Zscan4 (AB4340 1:200) primary antibodies and A-21206 488aRb and A-31571 647aMs secondaries. Samples were processed and analyzed as above. Significance was called using one-way ANOVA and Sidak's multiple comparisons test. The percentages of mESCs expressing the MERVL-EGFP reporter and the pluripotency-associated surface markers SSEA-1 and EpCAM were determined using live-cell flow cytometry using a BD FACS Canto. Similarly, cell-sorting of the MERVL-GFP+ and MERVL-GFP- was conducted with the help of the flow cytometry core facility at Weill Cornell. Live cells were run on the BD Influx on the FITC channel and gated based on high and low fluorescence, followed by cell sorting. Data analysis was done using FlowJo software. To follow the fate of 2CLCs and mESCs, 10,000 FACS-purified MERVL-GFP+ or MERVL-GFP- cells were plated in six replicates and allowed to grow in treatment media for 24 h. The percentage of cells in MERVL-GFP+ and MERVL-GFP- from each condition was determined by live flow using the BD Fortessa flow cytometer.

## Annexin V expression

MERVL-GFP mESCs were treated with DMSO or dTAG for 72 h. Cells were trypsinized and collected. Cells were washed twice with ice-cold PBS and resuspended in binding buffer (BD 556454). BD Pharmigen APC Annexin V (550474) was used to stain the cells as per the manufacturer's instructions. DAPI was added to the cells before analysis by flow cytometry on BDFortessa. Cells were categorized based on the following gating logic. Viable cells =

Annexin V-negative, DAPI-negative. Early apoptotic cells are Annexin V-positive and DAPI-negative. Late apoptotic cells are Annexin V-positive and DAPI-positive. Necrotic cells are Annexin V-negative, DAPI-positive.

## UNC0638 treatment and RT-qPCR

EHMT2-dTAG mESCs or MERVL-GFP mESCs were treated with DMSO, dTAG, or UNC0638 inhibitor for 24 h. The percentage of MERVL-positive cells in each condition was ascertained by flow cytometry. RNA was isolated from treated EHMT2-dTAG mESCs, as mentioned below. Reverse transcription of RNA from each sample was performed using the iScript kit (BioRad 1708841). qPCR was performed on cDNA samples in triplicate using PowerUp SYBR green PCR master mix (Thermo Fisher A25778) on an Applied Biosystems QuantStudio3. The primers used are in Dataset EV6.

## RNA isolation

Total RNA from cells was extracted using TRIzol (Invitrogen 15596018) and purified using the RNA Clean and Concentrator kit (Zymo Research ZR1014). RNA quality and quantity were checked before assays using a nanodrop or bioanalyzer.

## RNA sequencing

For the recovery experiment, EHMT-dTAG mESC clones (triplicates) were treated with DMSO or dTAG (200 nM) for 24 h, 7 days, and 15 days. After 7 days of dTAG treatment, the recovery group was allowed to grow in DMSO for 8 days. Cells were collected at the end of the treatment cycle, and RNA was extracted as described above. Following the manufacturer's instructions, 1 μg of RNA was then used to prepare libraries with the TruSeq Stranded mRNA Library Prep (Illumina #20020595) at the Weill Cornell Genomics Core. Libraries were sequenced on the NovaSeq 6000 S4 flow cell at a read length of PE 2 × 100. For the MERVL reporter experiments, MERVL-GFP mESCs were treated in triplicate with DMSO or dTAG (200 nM) for 72 h and sorted into GFP⁻ and GFP⁺ populations, as described in flow cytometry. In total, 200,000 cells were collected, and RNA was isolated using TriZol, as described above. Low-input RNA libraries were prepared by Novogene and sequenced at PE x150 read length on a Novaseq 6000. For the DPPA4-KO experiment, WTJ1 and DPPA4KO mESCs ($n = 3$ clonal lines) were treated with DMSO or UNC0638 for 48 h. RNA was isolated as described above. RNA libraries (polyA enrichment) were prepared by Novogene and sequenced on a Novaseq 6000 with a PE ×150 read length.

## Immunofluorescence

MERVL-GFP mESCs treated with DMSO or dTAG for 24 h were washed and fixed in 4% formaldehyde for 10 min, blocked in blocking buffer (PBS−/− with 1% BSA, 0.1% Triton-X-100, and 3% donkey serum (Sigma D9663) and stained with ZSCAN4 antibody (AB4340) at 1:150 for 2 h. Cells were washed three times with PBS −/− containing 0.1% Triton X-100 (PBST) and then incubated in Donkey anti-Rabbit Alexa Fluor 555 secondary antibody at a dilution of 1:1000 for 1 h. Cells were washed thrice with PBST.

DAPI (300 nM) was added for 5 min for nuclear staining. Images were taken on a Nikon fluorescent microscope. Images were quantified using ImageJ.

## Colony-forming assay

MERVL-GFP mESCs were treated with DMSO or dTAG for 72 h and sorted by flow cytometry as described above into GFP+ (2CLC) and GFP- (mESC) populations. Single cells were seeded into two 96-well plates per treatment group. Cells were allowed to grow in mESC media for 6 days. The number of colonies per plate was then counted. Alternatively, 1000 sorted cells (2CLC and mESC) were plated into each well of a six-well plate and allowed to grow in either DMSO or dTAG. Cells were allowed to grow for 5 days, after which alkaline phosphatase staining was performed using the Vector Red Alkaline Phosphatase Substrate Kit (SK-5100) according to the manufacturer's instructions, and images were captured. The number of colonies was counted using ImageJ.

## ATAC-Seq

ATAC-seq was performed as previously described (Buenrostro et al, 2015) with some modifications. Briefly, cells from two independent clones (EHMT2-dTAG mESCs) were treated with DMSO for 24 h. Cells were trypsinized, and 50,000 cells per replicate were washed with 50 μL cold 1× PBS followed by 50 μL lysis buffer (10 mM Tris-HCl, pH 7.4, 3 mM MgCl₂, 10 mM NaCl, 0.2% (v/v) IGEPAL CA-630) to isolate nuclei. Nuclei were pelleted by centrifuging for 10 min at 800 × g at 4 °C, and 50 μL of transposition reaction mix (25 μL TD buffer, 2.5 μL Tn5 transposase, and 22.5 μL ddH₂O) was added. Reagents from the Nextera DNA library Preparation Kit (Illumina #FC-121–103) were used. Samples were incubated at 37 °C for 30 min. DNA was isolated using the ZYMO Kit (D4014). ATAC-seq libraries were generated using NEBNext High-Fidelity 2× PCR Master Mix (NEB, #M0541), with each sample assigned a unique barcode and a universal primer. The optimal cycle number for each sample was determined by qPCR. Samples were size-selected (0.55×–1.5×) using SPRIselect beads (Beckman Coulter, B23317). Libraries were assessed with an Agilent Bioanalyzer. Libraries were sequenced on an Illumina Nova-Seq 6000 platform with 100 bp paired-end reads.

## ChIP-Seq

Two independent clones (EHMT2-dTAG mESCs) were treated with DMSO 24 h before ChIP-seq. ChIP was performed as previously described (Pelham-Webb et al, 2021) with some modifications. Briefly, 30 million cells per replicate for each condition were double-crosslinked by first incubating with 2 mM DSG (Sigma 80424-5MG-F) for 50 min, followed by 1% formaldehyde at RT for 10 min, and then quenched with 125 mM glycine for 5 min at RT. Cells were resuspended in 300 ml lysis buffer (10 mM Tris, pH 8, 1 mM EDTA, 0.5% SDS) and sonicated in a Pico bioruptor for 15-30 cycles and then centrifuged for 10 min at 4 °C at 17,000 × g. Supernatants were precleared for 1 h with 20 μL of Protein A Dynabeads (Thermo Scientific 10-001-D) per sample. 5% of each sample was removed and frozen as an input sample. Samples were diluted 5 times with dilution buffer (0.01% SDS, 1.1% Triton,1.2 mM EDTA,16.7 mM Tris, pH 8, 167 mM NaCl) and

incubated with HA antibody (5 μg/30 million total cells) (CST 3724) O/N with rotation at 4 °C. The next day, protein A Dynabeads pre-blocked with 1 mg/mL BSA protein were added to each sample (30 μL Dynabeads per sample). Samples were incubated for 3.5 h at 4 °C. Subsequent steps were carried out as previously described. According to the manufacturer's instructions, 15 ng of immunoprecipitated DNA and input were used for library amplification with the KAPA Hyper Prep kit (KK8502). Libraries were sequenced on an Illumina Novaseq 6000 platform with 100 bp paired-end reads. In a separate experiment, EHMT2-dTAG mESCs were treated with DMSO or dTAG for 24 h before ChIP-seq as described above, using HA antibody (ab9110)(4 μg/30 M cells total).

## ChIP-EXO

Cultures of EHMT2-dTAG mESCs were treated with DMSO for 24 h. Cells were collected by trypsinization for 3 min and were washed with PBS, followed by centrifugation (500 × g, 5 min, 4 °C). 10 M cells were used per replicate. The pellet was resuspended to a concentration of 1 million cells per mL in PBS (RT). Cells were crosslinked with 1% formaldehyde at RT for 10 min on a platform shaker. Crosslinking was quenched with 125 mM glycine for 5 min at RT on a platform shaker. Crosslinked cells were centrifuged (500 × g, 5 min, 4 °C) and were washed twice with ice-cold PBS. After the final wash and centrifuge, the pellet was snap-frozen before extraction. Frozen cell pellets were processed by the Cornell Epigenomics Core Facility as described previously in the ChIP-exo 5.0 protocol (Rossi Lai and Pugh, 2018). Anti-HA antibody (Abcam 9110) was used for o/n chromatin immunoprecipitation at 4 °C. Immunoprecipitated material was processed as described (Rossi Lai and Pugh, 2018).

## ULI-NChIP

A total of 100,000 cells were used as input for bulk mESC ULI-NChIP, with two independent replicates. For the 2CLC and mESC-sorted cells, three independent experiments were conducted using one clone. MERVL-GFP mESCs were treated with DMSO or dTAG-13 for 72 h in the presence of TTNBP (TT; 0.2 M) to reduce the length of sorting time required to obtain sufficient numbers of viable MERVL+ cells. Cells were sorted based on GFP expression, as described in the "Flow cytometry" section. ULI-NChIP was conducted following a published procedure (Brind'Amour et al, 2015). Briefly, frozen cell pellets were resuspended in Sigma EZ Nuclei Isolation Lysis Buffer (NUC101) containing 1× protease inhibitor cocktail (Roche 04693132001) and 1 mM PMSF (TF 36978), and then digested with MNase (NEB M0247). The digested chromatin was rotated for 1 h at 4 °C, followed by pre-clearing using Protein A/G beads (Dynabeads, Life Technologies #1006D), and 10% then saved as an input control. In total, 1 μg per 100k cells of H3K9me2 antibody (ab1220) was added to Protein A/G beads for 3 h at 4 °C to form the antibody complex. Precleared chromatin was added to the antibody complex and rotated at 4 °C overnight. The antibody-bound chromatin was then eluted from the magnetic beads using 100 mM NaHCO₃ and 1% SDS at 65 °C for 1 h and purified alongside the input chromatin. Libraries of the eluted and input chromatin were prepared using the ThruPLEX DNA-Seq (Takara, Cat #: R400675) following the manufacturer's instructions.

Libraries were sequenced on an Illumina NovaSeqXplus platform with 100 bp paired-end reads.

## CUT&RUN

MERVL-GFP mESCs (in duplicates) were treated with dTAG-13 for 72 h and sorted based on GFP as mentioned above in flow cytometry. CUT&RUN against DPPA4 was performed on 17,000–50,000 cells, as previously described (Ee et al, 2025; Skene Henikoff and Henikoff, 2018). Live cells were sorted based on GFP fluorescence on an Influx Cell Sorter before performing CUT&RUN. BioMag Plus Concanavalin A beads (ConA beads, PolySciences NC1358578) were washed twice with cold bead activation buffer (20 mM HEPES pH 7.5, 1 mM MnCl₂, 10 mM KCl, 1 mM CaCl₂). Sorted cells were washed twice in wash buffer (20 mM HEPES pH 7.5, 150 mM NaCl, 0.5 mM Spermidine (Acros AC132740050), ½ Protease Inhibitor tablet). At room temperature, 20 ml of activated ConA beads per sample were bound to the cells in wash buffer for 10 min. Samples were incubated overnight with gentle rocking at 4 °C with 1:100 of anti-DPPA4 (AF3730) or 1:100 anti-IgG (EpiCypher 130042) in antibody buffer (Wash Buffer with 2 mM EDTA and 0.01% Digitonin (Millipore-Sigma 300410)). Samples were washed twice with a wash buffer containing 0.01% Digitonin. ProteinA/G MNase (EpiCypher) was then bound to samples for 1 h at 4 °C. Protein A/G MNase was diluted in 50 μL of antibody buffer. Samples were washed twice in Digitonin Buffer, and 1 mL 100 mM CaCl₂ was added to activate MNase digestion. After two hours of incubation, MNase digestion was quenched using 33 mL of STOP Buffer (340 mM NaCl, 20 mM EDTA, 4 mM EGTA, 50 μg/mL RNase A, 50 μg/mL glycogen, 0.015 ng/mL E. coli spike-in) and the samples were incubated at 37 °C for 20 min. Digested fragments were isolated by centrifuging samples for 5 min at 16,000 × g, binding ConA beads to a magnet for 2 min, and then saving the supernatant. DNA was purified by adding 0.1% SDS and 5 μg Proteinase K for 10 min at 70 °C, followed by phenol-chloroform extraction and precipitation in 100% ethanol at −80 °C overnight. Pellets were washed in 100% ethanol and then resuspended in 12 mL of nuclease-free water. CUT&RUN libraries were prepared using the ThruPLEX DNA-Seq (Takara, Cat #: R400675) and Unique Dual Index (Takara #R400665) kits according to the manufacturer's instructions, except for the amplification step. After adding indexes, amplification cycles were performed using a shortened annealing/extension time (67 °C, 10 s) to enrich for small fragments, as previously recommended (Skene Henikoff and Henikoff, 2018). Libraries were size-selected with 1.5× volume SPRI beads (Beckman Coulter B23317) and sequenced at the Weill Cornell Genomics Resources Core Facility on an Illumina NovaSeq 6000 (PE-100, 30 million reads per sample).

## Data processing and analysis

All genomics datasets were processed using custom snakemake (v6.6.1) pipelines (31).

## RNA-seq

### Initial processing and explanation of the STAR multi-command
Technical replicates were merged using zcat. Reads were trimmed to remove adapter sequences using Trim Galore (v0.6.7) with

parameters: --phred33 --quality 0 --stringency 10 --length 20. Trimmed reads were mapped using STAR (v2.7.10) (Dobin et al, 2013). STAR alignment was performed using a previously described strategy developed to map multimapping reads to the best location in the genome (hereafter, "STAR multi") (Teissandier et al, 2019) using parameters: --readFilesCommand zcat --runThreadN 10 --outSAMtype BAM SortedByCoordinate --outFilterMultimapN-max 5000 --outSAMmultNmax 1 --outFilterMismatchNmax 3 --outMultimapperOrder Random --winAnchorMultimapNmax 5000 --alignEndsType EndToEnd --alignIntronMax 1 --alignMa-tesGapMax 350 --seedSearchStartLmax 30 --alignTranscriptsPer-ReadNmax 30000 --alignWindowsPerReadNmax 30000 --alignTranscriptsPerWindowNmax 300 --seedPerReadNmax 3000 --seedPerWindowNmax 300 --seedNoneLociPerWindow 1000 --alignSJoverhangMin 999 --alignSJDBoverhangMin 999. A detailed description of the modified STAR command is provided below. TE Counts (v2.2.3) (Jin et al, 2015b) was used to assign mapped reads to protein-coding and repeat genes and repeats with default parameters. GTFs from GENCODE (vM25) (Frankish et al, 2019) and a repeat file from TE Counts were used. Bigwigs were made from bed files using deeptools (v3.5.1) (Ramirez et al, 2014) using parameters: -p 10 --binSize 10 --ignoreForNormalization chrX chrM --normalizeUsing RPGC --effectiveGenomeSize 2652783500 --extendReads 200 --ignoreDuplicates.

**Explanation of STAR multi-command**

| Flag | Value | Description |
| --- | --- | --- |
| outFilterMultimapNmax | 5000 | The max number of places in the genome STAR will allow a read to align. Default is 10. |
| outSAMmultNmax | 1 | The number of entries STAR will write for each multi-mapper. Top scoring alignments are outputted first. |
| outFilterMismatchNmax | 3 | Number of mismatches to allow to be outputted. Default is 10 |
| outMultimapperOrder | Random | STAR should output the order of multimapping reads in random order. Without runRNGseed set alignments will vary between runs. |
| winAnchorMultimapNmax | 5000 | The number of loci that anchors can map to. winAnchorMultimapNmax must be >= outFilterMultimapNmax. Default is 50. |
| alignEndsType | EndToEnd | Turn off soft-clipping of reads. |
| alignIntronMax | 1 | Maximum intron size. Default is 0. |
| alignMatesGapMax | 350 | Maximum gap between two mates. Default is 0. |
| seedSearchStartLmax | 30 | Search start point through the read. The read is split into pieces <= this value. Default is 50. |
| alignTranscriptsPerReadNmax | 30,000 | Max number of different alignments per read to consider. Default is 100 |
| alignWindowsPerReadNmax | 30,000 | The max number of windows per read. Default is 100 |
| alignTranscriptsPerWindowNmax | 300 | The maximum number of transcripts per window. Default is 100 |
| seedPerReadNmax | 3000 | Max number of seeds per read. Default is 1000. |
| seedPerWindowNmax | 300 | Max number of seeds per window. Default is 50. |
| seedNoneLociPerWindow | 1000 | Max number of one seed loci per window. Default is 10 |
| alignSJoverhangMin | 999 | Minimum overhang for spliced alignments. Default is 5 |
| alignSJDBoverhangMin | 999 | Minimum overhang for annotated junctions. Default is 1 |

DESeq2 was used to identify differentially expressed genes (Love Huber and Anders, 2014). Differentially expressed genes were defined as genes with an adjusted $P$ value < 0.05 and log2 fold change (LFC) ≥ 1. Expressed genes were defined as genes that remained after DESeq2 performed independent filtering to remove genes with low counts ($P$-adj == "NA"). Normalized counts were generated using the counts function in DESeq2. For principal component analysis (PCA), the data were transformed using the vst function from DESeq2. The top 500 most variable genes were input into the prcomp function in R. For k-medoid clustering, normal-ized counts were z-scored, and a distance matrix was generated using the distNumeric function from the kmed package (Budiaji, 2022) with Manhattan weighted range (mrw). All genes were first assigned to a TAD to identify clusters of genes for ECORD analysis (Di Giammartino et al, 2019). Gene lists were then filtered to all differentially expressed genes, and the number of DEGs that occurred in sequence along the linear genome was counted. Clusters were broken if: (1) the next DEG was changed in the opposite direction ($P$-adj <0.05, no fold-cutoff), (2) the next gene was in a different TAD. After assigning each gene a cluster ID, the number of static genes was calculated by taking the full range of each cluster and intersecting it with the ranges of static genes. For subsequent analyses, "ECORDs" were defined as clusters with ≤50% static genes as well as ≥4 DEGS (LFC > 1) for the bulk dataset, and ≤50% static genes and ≥5 DEGs (LFC > 1) for the sorted-cell RNAseq dataset. For ChromHMM, gene promoters (TSS -2.5 kb/+0.5 kb) were used with a published, 100-state model (Vu and Ernst, 2023). After running, the values were rescaled to the range of 0 to 1 per column. For compartment analysis, the mean compartment signal across the entire gene body was calculated using a published dataset (Di Giammartino et al, 2019). For comparison with in vivo embryonic development, we used published FPKM values (Wu et al, 2016) and gene categories (Hu et al, 2020). To correct gene.ids in the in vivo dataset that did not match GENCODE annotations, the "select" function was used to

obtain aliases from the org.Mm.eg.db package, which was sufficient to correct nearly all inconsistencies. All plots were generated with ggplot2 (Wickham, 2010). For comparison to Zfp462 and Dppa4 DEGs, published datasets were reprocessed, and DEGs were called using the methods described above. The following published datasets were used: GSE176321 (zfp462 cl1, zfp426 cl2, WT ESC), GSE126920 (Dppa2 KO, Dppa4 KO, WT ESC).

## Gene Ontology (GO) analysis

Gene ontology analysis was done using ShinyGO (Ge Jung and Yao, 2020). Lists of genes were uploaded on the ShinyGO v0.81 server with the following settings: "Mouse" as the "Best matching species", "0.05" as the "P value cutoff (FDR)", and "25" as the '# of most significant terms to show'. All expressed genes (see above) in each dataset were used as background. For 24 h ECORD and non-ECORD, the terms obtained were collapsed using rrvgo (v3.19) (Sayols, 2023) and represented using R. For GO terms of overlapping DEGs, the most relevant terms were selected and represented using R.

## ChIP-seq and ChIP-Exo

### Initial processing

For repeat-aware alignment, reads were concatenated, trimmed, and aligned using STAR as for RNAseq above. For comparison to non-repeat aware alignment, ChIP and ChIP-Exo reads were also aligned using bowtie2 (v2.4.1) using parameters: --local --very-sensitive-local --no-unal –phred33 and then filtered using samtools only to include reads with a MAPQ score ≥ 5. PCR duplicates were removed using Picard (v2.26.0)(Tools) with parameters: VALIDATION_STRIN-GENCY = LENIENT REMOVE_DUPLICATES=true ASSUME_-SORTED=true. After duplicate removal, reads that overlapped ranges annotated as "High Signal Region" from the ENCODE Blacklist (mm10, v2) were removed using samtools (v1.14) (Li et al, 2009). Bam files were converted to bedpe files using bedtools (v2.30.0) (Quinlan and Hall, 2010) with parameters: -bedpe and then reformatted using the command: "sort -k 1,1 -k 2,2n | cut -f 1,2,6,7". Peaks were called on bed files using MACS2 (v2.2.6) (Zhang et al, 2008) using parameters: -f BEDPE --nomodel --seed 123 --keep-dup all -p 0.05. After peak calling with MACS2, peak files were filtered only to include peaks with q-value < 0.05 using a custom Python script. Bigwigs were made from bed files using deeptools (v3.5.1) (Ramirez et al, 2014) using parameters: --binSize 10 --ignoreForNormalization chrX chrM --normalizeUsing RPGC --effectiveGenomeSize 2652783500 --extend-Reads 200 --ignoreDuplicates --blackListFileName mm10-blacklist.v2_highSignalRegions.bed. Bigwigs were z-normalized per chromosome arm using a custom R script (v4.1.2). The following published datasets were also used: GSE137272 (Wiz), GSE158460 (Trim28), GSE177058 (Zfp462), GSE74112 (Oct4), GSE95517 (Dux), GSE113429 (H3K27ac, Klf4), GSE126921 (Dppa2, Dppa4).

ULI-NChIP reads were processed identically to ChIP-seq reads through to the generation of bigwigs. After generating RPGC normalized bigwigs, the bigwigs were re-binned at 10 kb resolution with a 5 kb sliding step using a custom R script as previously described (Au Yeung et al, 2019). Briefly, the mm10 genome was tiled into non-overlapping 5 kb bins. The bins were resized to 10 kb, anchored at the center, and the mean RPGC-normalized signal across the 10-kb window was calculated. Locations with zero signal

were ignored as these likely represent regions of the genome annotated as "High Signal Regions," which were removed during processing or unmappable regions. Bins were then resized to 5 KB before export as a BigWig. Consequently, each bin in the bigwig represents the mean across the 5-kb window/- 2.5 kb. Input-normalized tracks were generated by calculating log2(ChIP/Input) per bin. To identify H3K9me2 domains, regions of the genome with high H3K9me2 signal (defined as log2(ChIP/Input) >80th percentile) were extended up- and downstream until a region with low H3K9me2 enrichment was encountered (defined as log2(ChIP/Input) <30th percentile). The following packages were used: Genomic Ranges (Lawrence et al, 2013), plyranges (Lee Cook and Lawrence, 2019), rtracklayer (Lawrence Gentleman and Carey, 2009), and org.Mm.eg.db (Carlson, 2021).

For repeat analysis, the mm10_rmsk_TE.gtf repeat masker file from TE Transcripts (Jin et al, 2015a) was used. The fraction of binding sites that overlap repeats was determined at the base-pair level by calculating the number of bases within each peak that overlapped a repeat. This was done to make the values comparable to the whole genome fraction. For k-means clustering, union ranges of EHMT ChIP-seq and ChIP-Exo were generated, and the locations of the peak summits were annotated. The summit from the respective dataset was used for peaks called specifically to either ChIP-seq or ChIP-exo. For common peaks, the midpoint between the two summits was used. The signal from z-normalized bigwigs +/− 250 bp of peaks summits for all datasets, except for H3K9me2, was obtained using the getPlotSetArray function from seqplots (Stempor and Ahringer, 2016), updated to make it compatible with Bioconductor >v3.14 (https://github.com/cmuyehara/seqplots), using parameters: refgenome = "mm10", bin = 10, rm0 = F, ignore_strand = T, xanchored = 'mf', add_heatmap = T. For H3K9me2 ULI-NChIP, +/− 50k signal from the input-normalized bigwigs was used for clustering with the seqplots parameter modified to use a 1000 bp bin. Heatmaps from all datasets were clustered together using the k-means function in R after running set.seed(883). ChromHMM analysis on k-means clusters was performed on the union peak ranges, as described for RNA-seq above. To calculate the genomic distribution of binding sites in k-means clusters, the annotatePeakInBatch function from ChIPpeakAnno was used with parameters: PeakLocForDistance = "middle", output = "shortestDistance", FeatureLocForDistance = "middle", ignore.strand = T. To associate EHMT peaks with RNAseq, each expressed gene was assigned to the nearest EHMT peak (from the union dataset) in the same H3K9me2 domain. To calculate enrichment, a bootstrap approach was used, in which gene categories (e.g., 2CLC > mESC, static) were shuffled 1000 times. For PCA on ChIP and ChIP-Exo, raw counts from a union set of peaks were transformed using the vst function from DESeq2, and the top 500 most variable peaks were used as input to the pca function in R. For PCA on ULI-NChIP, the log₂(ChIP/Input) signal from all bins overlapping a union set of H3K9me2 domains was used. To estimate the proportion of the genome covered in H3K9me2 in DMSO and dTAG, a two-component gaussian model was fitted to the genome-wide distribution of log₂(ChIP/Input) signal with the mixtools package (Benaglia et al, 2009) and the lambda values for components 1 and 2 were used to estimate the proportion of the curve in either the "noise" or "signal" component, respectively. To

normalize between samples, sequencing-depth normalized bigwigs were subsequently scaled to the bigwig with the highest lambda value for component 2 ("signal").

## CUT&RUN

### Initial processing

Technical replicates were merged using zcat. Reads were trimmed using the bbduk.sh command from BBMap (Bushnell, 2014) using parameters: ktrim = r ref = adapters rcomp = t tpe = t tbo = t hdist =1 mink = 11. Trimmed reads were aligned using STAR, as described for RNA-seq above. Duplicate and blacklist removal were performed as for ChIP-seq above. Peaks were called on bed files using MACS2 (v2.2.6) (Zhang et al, 2008) with the following parameters: -f BEDPE --nomodel --seed 123 --keep-dup all. Bigwigs were z-normalized per chromosome arm using a custom R script.

Analysis was performed as described above for ChIP-seq and ChIP-Exo.

## ATACseq

ATAC-seq was processed similarly to CUT&RUN, as described above. To correct for the tn5 cut position, fragments were shifted by +4 bp and −5 bp for reads mapping to the + and − strands, respectively, using a custom awk script as recommended (Buenrostro et al, 2015).

## Expression of H3K9me2 regulators

Selected H3K9me2 regulators were represented in an expression heatmap constructed from a public in vivo dataset (Hu et al, 2020) and our in vitro MERVL-GFP RNA-seq data, as described above. Genes are represented as a log2 fold change over their respective levels in DMSO mESCs.

## Quantification and statistical analyses

Statistical analysis of Western blot, flow cytometry, and colony assays was done on PRISM (GraphPad Prism version 10.0.0, GraphPad Software, Boston, Massachusetts, USA, www.graphpad.com). Sample sizes were three or greater for flow cytometry, Western blot, and colony counting experiments, and two or greater for genomics experiments. No blinding was done during analysis. Specific tests and corrections applied are indicated in the respective figure legends and Dataset EV6, which also lists exact $P$ values.

## Data availability

RNA-seq, CUT&RUN, ChIP-seq, ChIP-exo, and ULI-NChIP-seq data have been deposited at Gene Expression Omnibus (GEO) under accession code GSE280606. The deposited data will be publicly available as of the publication date.

The source data of this paper are collected in the following database record: biostudies:S-SCDT-10_1038-S44319-025-00657-5.

## Peer review information

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

## Acknowledgements

We are grateful to Paul Zumbo, Friederike Dundar, and the Applied Bioinformatics Core at Weill Cornell Medicine for establishing STARmulti analysis, William Lai (Cornell Ithaca) and the Cornell EpiGenomics Core Facility for advice on ChIP-exo, Matthew Lorincz and Julie Brind'Amour for

advice on ULI-NChIP-seq, Charlene Raclot and Didier Trono for sharing DPPA2 and DPPA4 KO mESCs, and Wee Wei Tee for sharing gene expression data. We thank all members of the Apostolou and Stadtfeld labs for their valuable comments and support. CMU was supported by fellowships from the New York State Department of Health (NYSTEM training program) C32558GG and the National Heart, Lung, and Blood Institute (NHLBI) T32HL160520. MS was supported by grants from the NIH (R01GM145864), the Simons Foundation, the Tri-Institutional Stem Cell Initiative (Tri-SCI), and the Bohmfalk Charitable Trust. EA was funded by NIH (5R01GM138635 and 5RM1GM139738).

## Author contributions

**Kaushiki Chatterjee**: Formal analysis; Investigation; Visualization; Writing—review and editing. **Christopher Mitsuo Uyehara**: Data curation; Formal analysis; Visualization; Methodology; Writing—review and editing. **Kritika Kasliwal**: Formal analysis; Investigation; Methodology; Writing—review and editing. **Subhashini Madhuranath**: Resources. **Laurianne Scourzic**: Resources. **Alexander Polyzos**: Formal analysis; Methodology. **Effie Apostolou**: Supervision; Funding acquisition; Writing—review and editing. **Matthias Stadtfeld**: Conceptualization; Supervision; Funding acquisition; Writing—original draft; Writing—review and editing.

Source data underlying figure panels in this paper may have individual authorship assigned. Where available, figure panel/source data authorship is listed in the following database record: biostudies:S-SCDT-10_1038-S44319-025-00657-5.

## Disclosure and competing interests statement

The authors declare no competing interests.

