## [Peer Review File · EMBO Reports]

Coordinated repression of ZGA genes by methyltransferase EHMT2 via LINE1 regulatory elements

Matthias Stadtfeld, Kaushiki Chatterjee, Christopher Uyehara, Kritika Kasliwal, Subhashini Madhuranath, Laurianne Scourzic, Alexander Polyzos, and Effie Apostolou

Corresponding author(s): Matthias Stadtfeld (mas4011@med.cornell.edu) , Effie Apostolou (efa2001@med.cornell.edu)

Review Timeline:

Submission Date:	12th Mar 25
Editorial Decision:	19th Mar 25
Revision Received:	2nd Oct 25
Editorial Decision:	29th Oct 25
Revision Received:	7th Nov 25
Accepted:	18th Nov 25

Editor: Esther Schnapp

Transaction Report: The first round of review of this manuscript was performed at another journal.

Dear Matthias,

Thank you for the transfer of your manuscript with referee reports to EMBO reports, and for your revision plan.

I agree with the revisions you propose and would like to invite you to revise your manuscript with the understanding that the referee concerns must be fully addressed and their suggestions taken on board. Please address all referee concerns in a complete point-by-point response. Acceptance of the manuscript will depend on a positive outcome of a second round of review. It is EMBO reports policy to allow a single round of major revision only and acceptance or rejection of the manuscript will therefore depend on the completeness of your responses included in the next, final version of the manuscript.

We realize that it is difficult to revise to a specific deadline. In the interest of protecting the conceptual advance provided by the work, we recommend a revision within 3 months (19th Jun 2025). Please discuss the revision progress ahead of this time with the editor if you require more time to complete the revisions.

- 1) A data availability section providing access to data deposited in public databases is missing. If you have not deposited any data, please add a sentence to the data availability section that explains that.
- 2) Your manuscript contains statistics and error bars based on $n=2$. Please use scatter blots in these cases. No statistics should be calculated if $n=2$.

5) a complete author checklist, which you can download from our author guidelines <https://www.embopress.org/page/journal/14693178/authorguide>. Please insert information in the checklist that is also reflected in the manuscript. The completed author checklist will also be part of the RPF.

6) Please note that all corresponding authors are required to supply an ORCID ID for their name upon submission of a revised manuscript (<https://orcid.org/>). Please find instructions on how to link your ORCID ID to your account in our manuscript tracking system in our Author guidelines <https://www.embopress.org/page/journal/14693178/authorguide#authorshipguidelines>

7) Before submitting your revision, primary datasets produced in this study need to be deposited in an appropriate public database (see <https://www.embopress.org/page/journal/14693178/authorguide#datadeposition>). Please remember to provide a reviewer password if the datasets are not yet public. The accession numbers and database should be listed in a formal "Data Availability" section placed after Materials & Method (see also <https://www.embopress.org/page/journal/14693178/authorguide#datadeposition>). Please note that the Data Availability Section is restricted to new primary data that are part of this study. * Note - All links should resolve to a page where the data can be

accessed. *

12) All Materials and Methods need to be described in the main text using our 'Structured Methods' format, which is required for all research articles. According to this format, the Methods section includes a Reagents and Tools Table (listing key reagents, experimental models, software and relevant equipment and including their sources and relevant identifiers) followed by a Methods and Protocols section describing the methods using a step-by-step protocol format. The aim is to facilitate adoption of the methodologies across labs. More information on how to adhere to this format as well as a downloadable template (.docx) for the Reagents and Tools Table can be found in our author guidelines:

An example of a Method paper with Structured Methods can be found here: <https://www.embopress.org/doi/full/10.1038/s44320-024-00037-6#sec-4>

I look forward to seeing a revised form of your manuscript when it is ready.

Based on the reviewers' thoughtful comments, we conducted a substantial number of additional experiments and analyses and have added the following major new data to our study:

- Flow cytometry analysis to follow the fate of purified 2CLCs and mESCs depleted for EHMT2.
- RNA-seq analysis of DPPA4 KO mESCs exposed to EHMT2 inhibitor to identify DPPA-dependent and DPPA-independent EHMT2 target genes.
- HA ChIP-seq of dTAG-treated mESCs to validate the specificity of EHMT2 detection and document the degree of EHMT2 depletion from chromatin.
- ULI-nChIP to compare the levels of H3K9me2 in mESCs and spontaneously arising 2CLCs at ECORDs and other genomic regions.
- ULI-nChIP to compare the levels of H3K9me2 in EHMT2-depleted mESCs and 2CLCs to those of cells expressing physiological levels of EHMT2.

We describe the results of these experiments, as well as the additional analyses requested by the reviewers, in detail within the context of their specific comments listed below.

Reviewer #2 (Remarks to the Author):

In their paper, entitled Coordinated repression of totipotency-associated gene loci by histone methyltransferase EHMT2 through binding to LINE-1 regulatory elements, the authors study the role of EHMT2, an enzyme responsible for depositing the H3K9me2 histone PTM, in mouse embryonic stem cells. They find that the rapid and efficient degron-mediated depletion of EHMT2 – leading to the loss of the cognate PTM – results in reactivation of distinct gene sets, associated with opposing cell states. Notably, EHMT2 depletion reactivates genes typically expressed during early embryogenesis and in so-called 2-cell like cells (2CLCs), as well as genes promoting cell differentiation, as was previously reported. The most interesting finding is that coordinated reactivation of 2-CLC-genes can be explained by their physical localization, whereby they are organized in so-called ECORDs (for EHMT2-coordinately repressed domains), characterized by LINE-1 and MERVL element enrichment and H3K9me2-nucleation sites over LINE-1 sequences. Indeed, depletion of EHMT2, and consequently H3K9me2, significantly increases the population of 2-CLCs and their transcriptional programs. Conversely, EHMT2-regulated genes found outside of ECORDs are involved in differentiation and the authors proposed are regulated by Zfp462-mediated recruitment of EHMT2. Generally, this is a well-executed study, with appropriately designed experiments and balanced conclusions. The findings of this manuscript shed light on the seemingly paradoxical role of H3K9me2 in promoting cell fates of opposite directionality, and will certainly inspire new studies focusing on the coordinated ways transcriptional programs can be regulated in cis.

We thank the reviewer for their constructive assessment of our work.

There are three points we would like to raise, which would strengthen the overall quality of this MS:

1) Related to Figures 3 and S3: what is the ultimate fate of 2-CLCs which arise upon EHMT2 degradation? It is unclear whether they remain in that cell state, or they differentiate, or ultimately die. For example, showing they still are GFP+ and therefore expressing MERVL would help, especially as the authors claim 2CLCs are "locked" in their state but do not show any evidence supporting this. Further characterization and/or explanation would help clarify this matter.

The impact of EHMT2 depletion on the fate of 2CLCs is a crucial consideration. To address this question, we used flow cytometry to re-analyze the progeny of highly enriched (>95% purity; Fig.S3E) populations of EHMT2-dTAG 2CLCs (MERVL-GFP+ cells) established either under DMSO (wildtype levels of EHMT2) or dTAG (depleted for EHMT2) conditions. In these experiments, we took advantage of the fact that the mCherry reporter integrated into the EHMT2 degron allele is detectable in both 2CLCs and mESCs (data not shown), which allows us to capture all cell progenies, regardless of their identity. Re-analysis of purified cell populations 24 hours after sorting revealed that the majority of 2CLCs neither reverted to mESCs nor remained 2CLCs (Fig. 3L). This was observed in cells cultured in DMSO, but was even more pronounced in EHMT2-depleted cells. For example, 10,000 seeded 2CLCs yielded only about 100 mESCs and fewer than 70 2CLCs a day later (Fig. 3L). Flow cytometric analysis revealed higher levels of the apoptosis-associated surface marker Annexin V on 2CLCs than on mESCs, suggesting that 2CLCs are inherently unstable in standard mESC culture conditions. However, we did not observe differences in Annexin 5 expression between DMSO- and dTAG-treated cultures (Fig.S3N). Together, these observations suggest that EHMT2-depleted 2CLCs do not stably retain a 2CLC identity— at least under standard mESC conditions—but exhibit evidence of elevated molecular stress that may result in cell death via a non-apoptotic mechanism. However,

we cannot rule out other reasons why EHMT2-depleted 2CLCs might be lost from culture (e.g., impaired ability to attach to the cell culture substrate). We describe these new results in lines 345-352 of our revised manuscript, in which we have toned down our conclusions of a “locked” state

2) In Figure 4, the authors use ChIP-seq, ChIP-exo and ULI-NCHIP, and later in Figure 6, Cut&Run, for various chromatin profiling experiments. While the reasoning behind Cut&Run approach is justified, it is unclear to the reviewer why these different approaches are used beforehand. The low concordance between peaks called using ChIP-seq versus ChIP-EXO needs to be further elaborated on.

We apologize for not clearly stating our rationale for using both ChIP-seq and ChIP-exo to analyze the genome occupancy of EHMT2. We opted to do so since the chromatin binding of transcriptional regulators that do not directly engage with DNA, such as EHMT2, is inherently less stable and hence more difficult to determine. Employing two assays would give us more confidence in our results. The large number of predominantly weak “Exo-only” peaks we report may indicate that the favorable signal-to-noise ratio associated with ChIP-exo (Rhee and Pugh 2011, PMID 22153082) enables the detection of weak EHMT2 binding events (see Fig. S4E) that are masked by the higher background associated with ChIP-seq. Despite the overall weaker signal of ChIP-exo-specific peaks, the enrichment for H3K9me2 occupancy around them (Fig. 5A, Fig. S4E) further increases our confidence that we capture real EHMT2 binding events. Conversely, peaks exclusive to ChIP-seq might reflect EHMT2 being tethered further away from chromatin (for example, as part of a distinct repressive complex), which the more extensive fixation associated with our ChIP-seq protocol allowed us to capture. Importantly, our major conclusions regarding the molecular regulation of ECORDs through EHMT2 binding at LINE-1 elements are supported by both experiments (i.e., by peaks at LINE-1s detected by both ChIP-exo and ChIP-seq)(see, for example, Fig.S4G). Moreover, we have added a new HA ChIP-seq experiment to our revised manuscript, in which we demonstrate that 24 hours of dTAG treatment removes essentially all EHMT2 peaks (Fig. S4E), which supports the specificity of EHMT2 binding we report. We now further elaborate on the observed differences between ChIP-seq and ChIP-exo and describe this additional experiment in lines 370-384 of the revised manuscript.

3) Related to Figure 5, the authors indicate that ECORD 2-CLC genes upregulated upon EHMT2 KD could be dependent on Dppa2/4, as they were previously shown to be occupied by these factors in 2CLCs. However, this co-dependency was not formally tested in this MS. The authors used a chemical inhibitor of EHMT2 and performed qPCR on two candidate genes, but we would strongly recommend performing a concomitant depletion of EHMT2 and Dppa2/4 followed by RNA-sequencing and FACS analysis.

We agree that providing a broader assessment of how DPPA2/4 deficiency impacts the transcriptional changes caused by loss of EHMT2 would be informative, and we thank the reviewer for this suggestion. Since applying the EHMT2 inhibitor UNC0638 to DPPA4 KO mESCs is experimentally straightforward, we chose to pursue this approach. To do so, we initially compared the impact of EHMT2 depletion (via dTAG) and of EHMT2 inhibition (via UNC0638) using our MERVL reporter transgene. This revealed a similar increase in the frequency of 2CLCs compared to control cultures using both methods (Fig. S4J), indicating that the enzymatic activity of EHMT2 is critical in this context and that EHMT2 inhibition is therefore a suitable proxy for EHMT2 depletion. We then used RNA-seq analysis to globally compare the expression changes of EHMT2 target genes (as determined using culture of EHMT2-dTAG mESCs for 7 days in the presence of dTAG-13; see Fig.1) in DPPA4 KO mESCs cultured in the absence or presence of UNC0638. As controls for these experiments, we used J1 mESCs, the parental lines from which the DPPA4 KO cells were derived (De Iaco et al., 2019, PMID 30948459). These experiments showed that 2CLC/ZGA-associated EHMT2 DEGs, both within and outside of ECORDs, failed to be transcriptionally upregulated upon EHMT2 inhibition in the absence of DPPA4 (Fig. 5E and Table S2). In contrast, mESC-associated EHMT2 DEGs still experienced upregulation upon EHMT2 inhibition in the absence of DPPA4 (Fig.5E). These results – which we describe in lines 471-485 – document the general dependence of ZGA-associated DEGs repressed by EHMT2 on DPPA4 activity and further support our model of two distinct transcriptional programs controlled by EHMT2 in mESCs.

Minor points:

1) In line 239 there is a missing reference to figure S2C

Thank you for pointing this out. We believe the reviewer refers to Fig. S3C (the effect of passaging on the abundance of 2CLCs in mESC cultures). We have added this reference (line 253 in the revised manuscript).

2) In figure 5 the caption is not ordered accordingly to the text, please fix it
Thank you. We have corrected this.

Reviewer #3 (Remarks to the Author):

Chatterjee et al (NSMB-A50229)

Mouse embryonic stem cells (mESCs) and other naïve pluripotent stem cells are known to adopt a differentiation state (2-cell-like cells, 2CLCs) that mimics the zygotic genome activation (ZGA) stage of the early embryo development with totipotency. And from previous studies, histone H3K9 methyltransferase G9a/GLP is known to be one of the factors that negatively regulate the transition to such a state, and inhibition of G9a function increases the ratio of cells that acquire expression of 2CLC-specific ZGA-related genes (PMID22722858, PMID: 23735015). Therefore, in the present study, the authors used the G9a-dTAG dTAG-13-inducible depletion system in mESCs to dissect this system as a model for the regulation of totipotent-pluripotent differentiation in the early embryo, and to understand how G9a-mediated epigenomic regulation controls 2CLC gene expression in order to understand the regulation of early embryonic development in mammals at this stage. To identify induced 2CLC populations in mESC culture, the authors introduced a GFP reporter construct regulated by MERVL, one of the ZGA stage or 2CLC specifically activated genes/retroelements (Macfarlan et al 2012, Ishiuchi et al, 2015), then performed RNA-seq, H3K9me2 and G9a ChIP-seq analysis to investigate how G9a and G9a-mediated H3K9me2 contribute to the transcriptional regulation of 2CLC-specific genes. As a result, the authors found that G9a regulates transcription in two different modes; one is that the G9a-target gene regions (EHMT2 (G9a) Coordinately Repressed Domains (ECORDs)), which exists in clusters on same genome region, contains many 2CLC-specific ZGA-related genes, and G9a mainly targets LINE-1 in this region and expands H3K9me2 in the region, which eventually negatively regulates the expression of 2CLC-specific genes within the ECORDs by inhibiting the function of DPPA2/4 that contributes to the transcriptional activation of 2CLC-specific genes, the other regulates subsequent differentiation-specific gene expression directly by targeted via known G9a-associated ZFP462 and other DNA-binding factors. Finally, the authors conclude that "Our observations show that EHMT2 attenuates the bidirectional differentiation potential of mouse pluripotent stem cells and define molecular modes for locus-specific transcriptional repression by this essential histone methyltransferase."

Overall, the paper is well written and the proposed mechanism of transcriptional regulation of G9a, especially the mechanism by which G9a targets LINE-1 and indirectly regulates the expression of 2CLC-specific genes, would be potentially interesting if it is true. Furthermore, it is useful for the community in this field to have this unique perspective on the class of genes regulated by G9a in the ES and 2CLC states and to have a detailed summary of their properties. On the other hand, the fact that G9a regulates cluster genes itself is already a known finding (Lyons et al. 2014, PMID: 25437545; ; Myant et al. 2011, PMID: 21149390, PMID: 12130538), so does not seem novel in itself. And more important point is, the reviewer is still not fully convinced that the mechanism proposed by the authors is indeed the case, since the basic data, which is most important for the conclusions claimed in the current paper, is not shown. Furthermore, it is also not shown whether this regulation truly reflects an in vivo phenomenon. Therefore, the reviewer believes it is necessary to show some data, as pointed out below, to improve the QUALITY of this manuscript for publication.

We thank the reviewer for highlighting the general interest, usefulness, and importance of our findings, as well as for their constructive criticism.

Major comments,

1. H3K9me2 ChIP-seq analysis in naturally converted and dTAG-13 treated induced 2CLC populations

a) Collect (2-3% of) 2CLCs converted in normal ES culture by FACS sorting, perform H3K9me2 ULI-NChIP analysis on mESCs vs 2CLCs (due to the following paper, PMID: 39482357, 1×10^4 would be sufficient for H3K9me2 ULI-NChIP-seq analysis, or it would be also experimentally feasible to perform 1×10^5 order ULI-ChIP-seq analysis of 2CLC population as done by the authors in this manuscript), and to examine whether the levels of H3K9me2 are attenuated on ECORDs of 2CLC-specific genes in spontaneously 2CLC converted cells instead of G9a forced KD 2CLC cells. This exp is critical. If reduction of H3K9me2 over ECORDs or 2CLC-specific genes ECORDs is confirmed in spontaneous 2CLCs, it is better to check expression of potential molecules affecting H3K9me2 level between mESC and 2CLC, such as G9a, GLP, G9a/GLP targeting molecules and H3K9 demethylases.

We employed a two-step approach to quantify H3K9me2 levels in mESCs and 2CLCs and assess the impact of EHMT2 depletion on this chromatin mark. First, we used intracellular flow cytometry to compare the overall abundance of H3K9me2 across conditions. This revealed similar levels of H3K9me2 in mESC-enriched (ZSCAN4-) and 2CLCs-enriched (ZSCAN4+) populations in DMSO culture (Fig.S7D), demonstrating that the spontaneous conversion of mESCs into 2CLCs is not associated with a global reduction in H3K9me2

levels. As suggested by the reviewer, we then performed H3K9me2 ULI-nChIP on FACS-purified 2CLCs (MERVL+) and mESCs (MERVL-), using 100,000 cells in biological duplicate. Consistent with the flow cytometry results, ULI-nChIP showed no pronounced drop in global H3K9me2 levels in 2CLCs compared to mESCs (Fig.S7C). Notably, both flow cytometry and ULI-nChIP readily detected the loss of H3K9me2 triggered by EHMT2 depletion (Fig. S7C, D), demonstrating the assay's high sensitivity. We will discuss the impact of EHMT2 depletion on H3K9me2 in more detail in our reply to point 1b.

Analysis at the gene level showed that H3K9me2 levels at the TSS of 2CLC-associated ECORD DEGs were strongly elevated compared to those at mESC-associated DEGs and even more so compared to different groups of control genes (Fig.7A). Furthermore, we observed the highest levels of H3K9me2 at those ECORD genes that were further upregulated in 2CLCs lacking EHMT2 (“2CLC^{dTAG_UP}”; as defined in Fig.3I) (Fig.S7E). These observations are consistent with a repressive function of the H3K9me2 mark at ECORDs. To our (initial) surprise, promoter regions of 2CLC-associated ECORDs DEGs (Fig.7A) and MERVL elements (Fig.7B) showed only small reductions in H3K9me2 levels between mESCs and spontaneously-arising (DMSO conditions) 2CLCs. In contrast, EHMT2-bound LINEs within ECORDs – while retaining overall elevated H3K9me2 levels in both cell types – showed pronounced differences in the pattern of H3K9me2 between mESCs and 2CLCs (Fig.7D). Specifically, in 2CLCs we observed a local reduction of the H3K9me2 signal at the center of the EHMT2 peak in EHMT2-bound LINEs (as determined in bulk cultures and hence likely reflecting the binding status in mESCs)(Fig.7D). This observation suggests local chromatin remodeling (e.g. loss of nucleosomes or loss of repressive histones), consistent with a change of ECORD LINEs from repressed to active regions with potential enhancer activity during the mESC-to-2CLC conversion. Of note, an enhancer function of LINE-1 elements in totipotent cells has also been recently suggested by Li et al. 2024, PMID 38849613. Together, our new findings – which we describe in lines 543-557 and discuss in lines 626-634 – are consistent with the notion that the spontaneous conversion of mESCs to 2CLCs is not associated with a dramatic loss of H3K9me2 across ECORDs, but rather with a specific remodeling of this mark at candidate distal gene regulatory elements such as LINEs.

Notably, two-cell stage mouse embryos have been reported to lack H3K9me2 (Zylicz et al., 2018, PMID 29745895; Deng et al., 2020, PMID 31746724), suggesting that the retention of this repressive mark at ECORDs and elsewhere in the genome in 2CLCs represents a molecular difference between spontaneously arising 2CLCs and early mouse embryos. To further explore these differences, we found it informative – as suggested by the reviewer – to evaluate the expression levels of core components of the H3K9 methylation machinery using public in vivo datasets (Hu et al., 2020, PMID 31932739) and our RNA-seq data. This analysis (shown in Fig.S6A) revealed pronounced differences in the expression of genes encoding methyltransferases (EHMT2, SETDB1) and demethylases (PHF2/8, KDM3A/B, JMJD1C, KDM4A-C) of H3K9 between 2-cell embryos and mESCs, but not between 2CLCs and mESCs. Even though clearly more work is required, this finding further defines molecular differences between 2-cell embryos and spontaneously derived 2CLCs, and, together with the transcriptional changes triggered in 2CLCs upon EHMT2 loss (Fig. 3I,J), supports the notion that EHMT2 depletion renders 2CLCs closer to ZGA-stage mouse embryos.

1b) The reviewer also highly requests H3K9me2 ULI-NChIP-seq analysis of ES and 2CLC populations after 7 days of dTAG-13 treatment. Since H3K9me2 is regulated by 5 H3K9 methyltransferases (PMID: 33986449) and LINE-1 transcription is known to be repressed by SETDB1 and SUV39H, it is possible that H3K9me2 is not attenuated on LINE-1s by G9a KD.

We have now conducted H3K9me2 ULI-nChIP in EHMT2-depleted 2CLCs and mESCs. These experiments revealed a significant drop in global H3K9me2 levels in both cell types upon dTAG treatment (Fig. S7C), which was also evident by flow cytometric analysis (Fig. S7D), confirming that the maintenance of this chromatin mark in both cell types depends on EHMT2. At the level of specific loci, we observed reduced H3K9me2 at the TSS of ECORD DEGs (Fig.7A) and ECORD MERVLs (Fig.7B) in both mESCs and 2CLCs lacking EHMT2. This reduction was more pronounced in mESCs, which might be due to the rapid cell cycle and associated passive dilution of chromatin marks in these cells. Interestingly, we observed a more complex situation at ECORD LINEs. Thus, while mESCs showed strongly reduced H3K9me2 at LINEs and other TEs upon EHMT2 depletion (Fig.7D; upper row), 2CLCs cultured in dTAG showed similar levels of H3K9me2 at LINEs close to the site of EHMT2 binding (see also our response to point 1a) than 2CLCs cultured in DMSO (Fig.7D; lower row). In contrast, regions further away from the EHMT2 peak exhibited progressively reduced H3K9me2 levels in EHMT2-depleted 2CLCs (Fig.7D; lower row). We interpret these results as supporting a requirement for EHMT2 to maintain repressive domains in both mESCs and 2CLCs via H3K9me2 spreading (including at ECORDs), while also suggesting local compensation at LINEs in 2CLCs but not in mESCs by other H3K9 methyltransferases. In agreement with this conclusion, we found that

TRIM28 – a recruiter of different chromatin repressors, including SETDB1 – is also present at EHMT2-bound LINEs (Fig.5A). We describe this observation in lines 555-557 and 628-632 of the revised manuscript.

2. G9a ChIP-seq analysis in dTAG-13 treated ES cells

a) Since G9a ChIP-seq analysis in ES is already reported (PMID: 24389103), it is nice to check if G9a is also enriched in LINE-1 in their data set.

We have re-analyzed the G9A ChIP-seq data in the paper referenced by the reviewer using our analysis pipeline, which improves the detection of occupancy over repetitive elements. This documented an overall poorer signal-to-noise ratio in the published study, as indicated by a lower FRiP (Fraction of Reads in Peaks) score (Supporting Figure 1A). Nevertheless, our analysis confirmed an elevated G9A signal at LINE-1 elements in the public ChIP-seq data and a high correlation with our ChIP experiments (Supporting Figure 1B). These observations provide additional, independent support for our findings and highlight the current scarcity of high-quality public G9A ChIP-seq datasets.

Even if such a behavior of G9a is not confirmed in the reported one, the finding of G9a enrichment in LINE-1 is the most novel result in this paper, and therefore, including the verification of specificity, G9a ChIP-seq in ES of dTAG-13+ 7days (either ChIP-seq or ChIP-EXO) should be performed to clearly demonstrate the

Supporting Figure 1. A) Fraction of Reads in Peaks (FRiP) score of the EHMT2 ChIP-seq in our study and in Mozzetta et al., 2014 (24389103). **B)** Analysis of public G9A ChIP-seq data. **B)** Correlation plots of ChIP signal strength at L1Mds elements in our and published EHMT2 ChIP-seq data (r = Spearman coefficient).

specificity of the G9a peaks

shown in this study.

We agree with the importance of documenting the reliability of EHMT2 detection in our ChIP approaches. We have therefore conducted another round of HA ChIP-seq in mESCs, including dTAG-treated cells. This experiment confirmed EHMT2 binding to peaks identified in our original ChIP-seq experiments and revealed a complete loss of signal at these sites upon EHMT2 depletion (Fig. S4E). This observation supports both the effectiveness of our dTAG allele in depleting EHMT2 from chromatin and the reliability of the EHMT2 peaks we have identified. We describe these new findings in lines 380-384 of the revised manuscript.

b) Furthermore, the fact that only 20% of the G9a peaks were overlapped in the two ChIP-seq methodologies (conventional method and ChIP-Exo), is not clearly explained why this is the case. The reviewer felt that a more detailed description of the characteristics of the peaks commonly and uniquely observed in each methodology is needed.

We believe that the more favorable signal-to-noise ratio of ChIP-exo (Rhee and Pugh 2011, PMID22153082) might explain the (predominantly weak) EHMT2 peaks only detected by this method (see Fig.S4E). Despite the overall weaker signal of ChIP-exo-specific peaks, the enrichment for H3K9me2 occupancy around them (Fig. 5A and Fig. S4E) further increases our confidence that we have captured real EHMT2 binding events. On the other hand, peaks exclusive to ChIP-seq might reflect EHMT2 being tethered further away from chromatin (for example, as part of a distinct repressive complex), which the more extensive fixation associated with our ChIP-seq protocol allowed us to capture. As already mentioned above, all these peaks are no longer detected in the dTAG sample, further supporting the validity of our experiments. We now elaborate on the observed differences between ChIP-seq and ChIP-exo in lines 370-380 of the revised manuscript. We have summarized the key features of the different peak categories in a text box adjacent to Fig. 5A.

Related to this, what happens if the analysis of Fig. 4A using only Common peaks?

This is an interesting question. Restricting analysis to peaks only detected in both ChIP-exo and ChIP-seq confirmed that peaks detected with both methods predominantly (>90%; see updated Fig.4A) bind to TEs. This confirms that the vast majority of EHMT2 peaks detected with both methods (“high confidence peaks”) occur at repeat elements, specifically at LINE (Fig. S4G).

3. Whether G9a really negatively regulates the expression of ECORDs genes by their hypothesis? Thus, 2CLC-specific ZGA-related genes are negatively regulated by G9a through targeting LINE-1 primarily and then suppressing transcriptional activation by DPPA2/4 by spreading H3K9me2 from it. The reviewer requests to validate their working hypothesis by any functional assay. For example, since DPPA2/4 directly regulates DUX expression positively (PMID: 30692203), it might be a good idea to do a reporter assay with the DUX promoter. If the DUX promoter can induce uniform expression of the reporter gene in ES cells, it is possible to test whether connecting LINE-1 to the construct causes transcription to be repressed, and if so, whether the repression is dependent on G9a or not. Also, it is interesting to check whether DPPA2/4 binding to the DUX promoter region is affected or not. Alternatively, deletion of any LINE-1 element targeted by G9a in ECORDs with 2CLC-specific ZGA-related genes (observed peaks in the G9a ChIP-seq analysis) in the G9a degron mESCs and examine the impact of this LINE-1 element depletion on H3K9me2 level in this region (in particular, how much 2CLC-specific ZGA-related genes in this ECORD will be re-silenced or DPPA2/4 binding will be affected after removal of dTAG-13 or G9a inhibitor). If results support the proposed authors' model, the quality of this paper will be highly enhanced.

We agree that the experiments suggested by the reviewer would be interesting and potentially informative; however, we consider them beyond the scope of our current study. We now mention potential alternative and future experiments to further characterize the biological functions of candidate LINE-1 elements in lines 644-646 of the revised manuscript.

4. Related to comment 1-a), if the 2CLC induction rate is increased by decreasing the amount or activity of the G9a/GLP complex or by increasing H3K9 demethylase activity, then overexpression of G9a/GLP should suppress this effect. It would be a good idea to overexpress G9a/GLP in ES to see what happens to the 2CLC incidence in such situations (even if there is no effect, it can be argued that G9a contributes to the regulation of expression of 2CLC-specific ZGA-related genes by mechanisms other than such).

While undoubtedly an interesting experiment, we decided against attempting to set up an ectopic expression system in the context of this study due to intrinsic difficulties associated with these approaches, including interpreting the consequences of superphysiological expression levels.

Minor comments

5. There is no clear statement, but is there any growth defect with dTAG-13 treatment? It would not be surprising if there is some growth defect.

We did not observe any evidence of an overt decrease in growth rate (for example, a less frequent need for passaging) during our long-term dTAG experiments. We added a statement to this effect to the revised manuscript (lines 150-152).

6. Fig. Legend of S6 is incorrect. Please revise it.

Thank you. We have corrected this.

7. Fig. 5 legend is not clear enough; labels C-F should be B-E; z-score and ChIP/input in A correspond to which columns?; C legend is not clear enough, what fractions are in each column, the reviewer could not understand from this explanation only; same for E, the legend says *Dppa2/4* KO, but the figure says *Dppa4* KO, also what kind of cell is EHMT-HA? There is no explanation at all.

Thank you for bringing these issues to our attention. We have made the following changes to the legends of Figure 5: 1) We have fixed the labeling; 2) We have clarified that z-score values in Fig. 5A apply to all columns but the H3K9me2 column, which shows ChIP/input values; 3) We have clarified that the UpSet plot in Fig. 5C below the graph defines the genes represented in each column (i.e., column 1 = ECORD genes downregulated in *Dppa2* or *Dppa4* KO, but not affected in *Zfp462* KO mESCs, etc.); 4) we have clarified that Fig. 5E shows data (now RNA-seq) obtained with *Dppa4* KO and parental J1 mESCs.

8. Which figures given for reproducibility of data between RNA-seq and ChIP-seq replicates? Correlation between replicate in RNA-seq, ChIP-seq should be included as suppl data.

We have now included PCA graphs of: A) RNA-seq with unsorted mESC cultures 24h and 7d after dTAG treatment (Fig.S1E); B) RNA-seq with sorted cell populations (Fig.S3F); HA-ChIP-seq and HA-ChIP-exo (Fig.S4A); DPPA4 CUT&RUN (Fig.S6B); and H3K9me2 ULI-nChIP of 2CLCs and mESCs (Fig.S7A).

Dear Matthias,

Thank you for the submission of your revised manuscript. We have now received the enclosed reports from the referees that were asked to assess it and I am happy to say that both support its publication now. Referee 1 has 2 minor suggestions that I would like you to address before we can proceed with the official acceptance of your manuscript.

A few editorial requests will also need to be addressed:

- Please reduce the number of keywords to 5.
 - The "Data and code availability" subheading needs to be corrected to "Data Availability Section". The specific URL for the GEO dataset GSE28060 needs to be provided in this section.
 - The conflict of interest subheading needs to be corrected to "Disclosure and Competing Interests Statement"
 - The format of the author list in the ms needs to be the same as in our online submission system: First Name Last Name (instead of Last Name first name initial); the other corr. author also needs her email address on the title page.
 - The author credits need to be removed from the ms file. All credits need to be entered during online ms submission.
 - DATA NOT SHOWN on page 9 needs to be removed as per journal policy. Please either show the data or rephrase.
 - The two funders provided in the Comments box in our online system (Tri-Institutional Stem Cell Initiative and Bohmfalk Charitable Trust) need to be removed from the box and entered as separate funders using the More Funders option.
 - In the figure callouts, "Supplemental" needs to be deleted.
 - There are 6 suppl. tables uploaded as separate Excel files. Each table needs to be updated to a Dataset so that you have separate Datasets EV1-EV6. The legends need to be removed from the ms file and each should be provided in its corresponding Excel file as a separate sheet/tab. The correct nomenclature (Dataset EV1-Dataset EV6) needs to be updated in all places: source file names, legends in each file, titles in the system, callouts in the ms.
 - A PDF with 7 suppl. figures is provided; the PDF needs to be clean (no color, highlighting, tracked changes, etc.); the correct title should be Appendix and the title page should only have a table of content with each Appendix item listed with its page number; the correct nomenclature should be Appendix Figure S1, etc. and the callouts in the ms need to be updated accordingly.
 - The Reagents & Tools TABLE needs to be removed from the ms file and uploaded separately and the table template from our guide to authors should be used.
 - Summary should be Abstract
 - Lead contact section should be removed from the ms
 - The manuscript sections should be in the following order: Title page - Abstract & Keywords - Introduction - Results - Discussion - Methods - Data Availability - Acknowledgments - Disclosure Statement & Competing Interests - References - Figure Legends - (Main Tables with legends if applicable) - Expanded View Figure Legends.
 - SUPPLEMENTAL INFORMATION TITLES AND LEGENDS needs to be removed from the ms
- *Figure Legends - Comments*
- Please note that the exact p values are not provided in the legends of figures 1B, 3C, E, K, L; 4D, E; 5E. Please provide exact p-values as reasonable.
 - Please indicate the statistical test used for data analysis in the legends of figures 1C, D
 - Please note that the box plots need to be defined in terms of minima, maxima, centre, bounds of box and whiskers, and percentile in the legends of figures 2B, C, G; 4D, 5E, 7A
 - Please note that information related to n is missing in the legends of figures 1C, D; 2B, C, G; 7A.

I would like to suggest one minor change to the abstract that needs to be written in present tense according to journal policy. Do you agree with this change :

We found that EHMT2 directly represses large clusters of co-regulated gene loci...
change to

We show here that EHMT2 directly represses large clusters of co-regulated gene loci...

EMBO press papers are accompanied online by A) a short (1-2 sentences) summary of the findings and their significance, B) 2-3 bullet points highlighting key results and C) a synopsis image that is exactly 550 pixels wide and 200-600 pixels high (the height is variable). The synopsis image should provide a sketch of the major findings, like a graphical abstract. Please note that text needs to be readable at the final size. Please send us this information along with the final manuscript.

Referee #1:

We have carefully examined the revised MS and the point-by-point answers from the authors.

We consider this to be a relevant and novel paper, and are overall supportive of its publication. We do not ask for any additional analyses or experiments.

That said, we would like to raise two minor points which could aid in improving the overall clarity and legibility of the manuscript.

1) Figures 3K, L are unintuitive. We suggest that the authors plot and display the ratio of the total number of seeded cells, rather than an absolute number (as its plotted now).

2) In the response to our point 1, the authors write: 'Together, these observations suggest that EHMT2-depleted 2CLCs do not stably retain a 2CLC identity- at least under standard mESC conditions -but exhibit evidence of elevated molecular stress that may result in cell death via a non-apoptotic mechanism'. We think this sentence should be included in the main text of the manuscript as it represents the actual phenotype they describe.

Referee #2:

The authors responded to most of the reviewer's critical comments. The reviewer minimally satisfied with there revision. So, no further comments I have.

Referee #1 (previously Reviewer #2):

We have carefully examined the revised MS and the point-by-point answers from the authors. We consider this to be a relevant and novel paper, and are overall supportive of its publication. We do not ask for any additional analyses or experiments.

We thank the reviewer for their positive and constructive feedback.

That said, we would like to raise two minor points which could aid in improving the overall clarity and legibility of the manuscript.

1) Figures 3K, L are unintuitive. We suggest that the authors plot and display the ratio of the total number of seeded cells, rather than an absolute number (as its plotted now).

We have changed the axis of these panels as suggested by the reviewer.

2) In the response to our point 1, the authors write: 'Together, these observations suggest that EHMT2-depleted 2CLCs do not stably retain a 2CLC identity- at least under standard mESC conditions -but exhibit evidence of elevated molecular stress that may result in cell death via a non-apoptotic mechanism'. We think this sentence should be included in the main text of the manuscript as it represents the actual phenotype they describe.

We have updated lines 353-356 of the manuscript in accordance with the reviewer's suggestion.

Referee #2 (previously Reviewer #3):

The authors responded to most of the reviewer's critical comments. The reviewer minimally satisfied with there revision. So, no further comments I have.

We thank the reviewer for supporting the publication of our work.

Matthias Stadtfeld
Weill Cornell Medicine
Medicine
413 E69th Street
Belfer Research Building, Room 1518
New York, New York 10021
United States

Dear Matthias,

I am very pleased to accept your manuscript for publication in the next available issue of EMBO reports. Thank you for your contribution to our journal.

Best,
Esther
